# Enhancing Robustness to Class-Conditional Distribution Shift in Long-Tailed Recognition

**Keliang Li**  *keliang.li@vipl.ict.ac.cn*
*Institute of Computing Technology, Chinese Academy of Sciences*
*University of Chinese Academy of Sciences*

**Hong Chang**  *changhong@ict.ac.cn*
*Institute of Computing Technology, Chinese Academy of Sciences*
*University of Chinese Academy of Sciences*

**Shiguang Shan**  *sgshan@ict.ac.cn*
*Institute of Computing Technology, Chinese Academy of Sciences*
*University of Chinese Academy of Sciences*

**Xilin Chen**  *xlchen@ict.ac.cn*
*Institute of Computing Technology, Chinese Academy of Sciences*
*University of Chinese Academy of Sciences*

**Reviewed on OpenReview:** *https://openreview.net/forum?id=n2gAD8Fdzk*

## Abstract

For long-tailed recognition problem, beyond imbalanced label distribution, unreliable empirical data distribution due to instance scarcity has recently emerged as a concern. It inevitably causes Class-Conditional Distribution (CCD) shift between training and test. Data augmentation and head-to-tail information transfer methods indirectly alleviate the problem by synthesizing novel examples but may remain biased. In this paper, we conduct a thorough study on the impact of CCD shift and propose Distributionally Robust Augmentation (DRA) to directly train models robust to the shift. DRA admits a novel generalization bound reflecting the benefit of distributional robustness to CCD shift for long-tailed recognition. Extensive experiments show DRA greatly improves existing re-balancing and data augmentation methods when cooperating with them. It also alleviates the recently discovered saddle-point issue, verifying its ability to achieve enhanced robustness.

## 1 Introduction

Visual recognition has recently achieved significant progress, driven by the development of deep neural networks (He et al., 2016) as well as manually balanced datasets (Russakovsky et al., 2015). However, real-world data usually exhibits long-tailed distribution over classes (Liu et al., 2019; Van Horn & Perona, 2017), leading to undesired estimation bias and poor generalization (Zhou et al., 2020; Cao et al., 2019; Kang et al., 2019). Recently, considerable efforts have been made to address imbalanced label distribution issue through re-balancing strategy. Two-stage methods (Samuel & Chechik, 2021; Cao et al., 2019; Zhou et al., 2020) and logit adjustment (Menon et al., 2020; Ren et al., 2020) have shown significant performance improvement over intuitive direct re-balancing (Zhang et al., 2021b), like re-sampling and re-weighting (Kang et al., 2019; Zhou et al., 2020), and become important components of recent solutions (Zhang et al., 2021a; Zhong et al., 2021). The rationality of re-balancing strategies (Cao et al., 2019; Park et al., 2021; Ren et al., 2020) has been theoretically proved (Menon et al., 2020), under the following assumption: the class distribution $P(y)$ shifts from training to test (usually class-imbalanced in training but class-balanced in test), while the *Class-Conditional Distribution* (CCD) $P(x|y)$ keeps consistent, i.e. $P_{train}(x|y) = P_{test}(x|y)$ (Menon et al., 2020; Ren et al., 2020).

Nevertheless, the consistent CCD assumption hardly holds in real-world long-tailed scenarios, attributed to instance sparsity. For tail classes where the samples are extremely scarce, estimating $P_{train}(x|y)$ by empirical CCD is unreliable at all and the shift between them cannot be ignored, as we demonstrate later in this paper. To address instance scarcity, some works (Kim et al., 2020; Zhou et al., 2022; Liu et al., 2019)

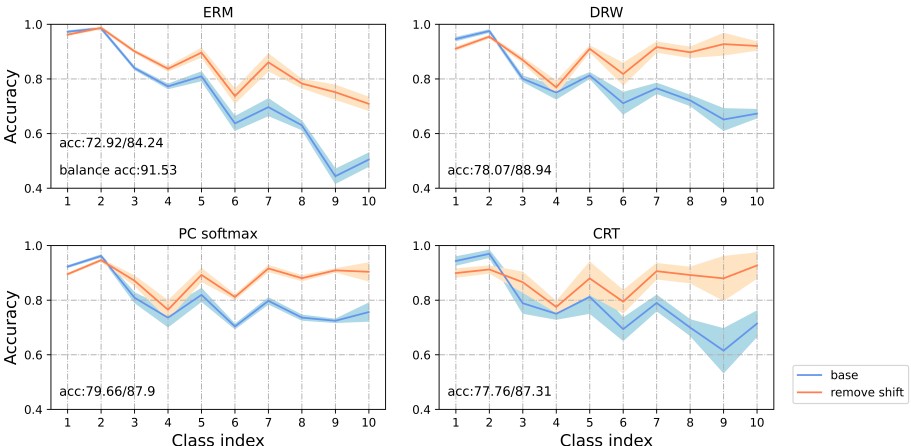

Figure 1: Accuracy on CIFAR10-LT (imbalance ratio: 100) with or without removing CCD shift. All methods (as introduced in Section 3.2) show significant improvement after removing shifts, especially for classes with fewer instances, verifying that the empirical CCD distributions of tail classes are more unreliable. Shaded regions show 95% CIs over 5 runs.

explore transferring information from head to enhance tail classes, e.g., by pasting image patches or mixing features (Park et al., 2022; Hong et al., 2022). However, head-to-tail transfer relies on the assumption (Chu et al., 2020; Kim et al., 2020) that (1) head class information can help to recover tail class distribution, and (2) class-generic and class-specific information can be decoupled and re-mixed to generate novel tail class samples. Actually, head-to-tail transfer cannot convincingly recover the real CCD. Even if class-generic information can be extracted perfectly, it may not be suitable to the tail classes e.g. the class-generic context (Park et al., 2022) shared by head classes, grassland, clearly contradicts the true distribution of tail-class animals that live in caves.

Other works (Xu et al., 2021; Wei et al., 2022; Zhong et al., 2021; Li et al., 2021; Ahn et al., 2023) dealing with instance scarcity employ data augmentation designed for balanced datasets to increase tail class diversity. However, there are concerns regarding these methods. Their target is generating novel instances instead of alleviating CCD shift directly, potentially leading to biased augmented examples. Moreover, existing data augmentation methods require specific modifications to adapt to long-tailed distribution setting (Li et al., 2021; Xu et al., 2021; Park et al., 2022). This complicates the joint use of data augmentation and re-balancing strategies, e.g. Mixup-based augmentation (Zhang et al., 2017) degrades margin-based re-balancing methods (Ren et al., 2020; Cao et al., 2019) in most cases (Xu et al., 2021; Park et al., 2022). Therefore, exploring data augmentation methods that can directly address CCD shift and cooperate with existing strategies is promising.

To this end, we first conduct an empirical study to quantify the impact that CCD shift has on long-tailed recognition. We verify that CCD shift is a critical factor limiting the performance of long-tail recognition, and highly correlated with the class cardinality, as in Figure 1. Moreover, our experiments explain the counter-intuitive phenomenon in state-of-the-art re-balancing methods and highlight the importance of instance-level discriminative data augmentation and logit adjustment methods in addressing CCD shift. Based on them, we design a novel data augmentation method, **D**istributionally **R**obust **A**ugmentation (**DRA**), that directly targets CCD shift and adapts to the current instance and training dynamics. Specifically, we train models to be more robust to unreliable empirical distributions rather than attempting to restore the underlying tail distribution, as expected in previous works (Hong et al., 2022; Park et al., 2022). We achieve this goal through Distributionally Robust Optimization (DRO) (Sinha et al., 2017; Lin et al., 2022) with a class-aware

uncertainty set, which allocates robustness based on the unreliability of each class. We further prove that, under mild conditions, our training objective bounds the balanced loss on real CCDs.

Our main contributions are highlighted as follows:

- We investigate CCD shift issue in long-tailed recognition quantitatively, and provide new insights from this view into main-stream re-balancing and data augmentation methods.

- We propose a theoretically sound method, Distributionally Robust Augmentation (DRA), to directly address instance scarcity problem and make models more robust to CCD shift.

- Extensive experiments on long-tailed classification benchmarks demonstrate that DRA can cooperate with and enhance existing re-balancing and data augmentation methods. DRA can also improve the confidence calibration (Guo et al., 2017) and alleviate saddle-point issue (Rangwani et al., 2022a).

## 2 Related Works

### 2.1 Re-balancing methods for long-tailed data

We review a broader scope of re-balancing methods that address the prior shift of $P(y)$. Intuitive direct re-sampling or re-weighting by frequencies is shown getting marginal improvement (Zhang et al., 2021b; Cui et al., 2019; Lin et al., 2017) while two-stage methods, e.g. deferring use of re-balancing strategies (e.g. DRS/DRW (Cao et al., 2019; Zhou et al., 2020; Park et al., 2021)) and decoupling methods which only re-balance the classifier (e.g. CRT, LWS (Kang et al., 2019)) overperform them significantly (Zhong et al., 2021; Zhang et al., 2021a). Another line introduces class-wise margin (Cao et al., 2019; Wu et al., 2021) or temperature (Ye et al., 2020) to cross-entropy loss. Logit adjustment (Menon et al., 2020) utilizes margin-based loss and post-hoc adjustment to modify logits, equivalent to PC softmax and Balanced softmax respectively (Hong et al., 2021; Ren et al., 2020), which has been proved consistent with balanced accuracy (Menon et al., 2020). Though two-stage methods and logit adjustment have become important components of recent solutions (Samuel & Chechik, 2021; Zhang et al., 2021a; Park et al., 2022; Hong et al., 2021; Zhou et al., 2023), there still remain questions to them, e.g. why deferring re-balancing is critical and why the consistent adjustment is sub-optimal. A concurrent work (Wang et al., 2023) sheds light on these questions by a systematical Fisher consistency analysis for re-weighted margin-based loss(Cao et al., 2019; Kini et al., 2021) while Section 3 will give explanations from the view of CCD shift. Additionally, re-balancing methods suffer from saddle points for loss of tail classes(Rangwani et al., 2022a), leading to poor generalization and instability to distribution shift(Dauphin et al., 2014). Rangwani et al. (2022a) and Zhou et al. (2023) use SAM technique(Foret et al., 2020) to improve re-balancing methods by allowing model to converge to flat minima. Surprisingly, our proposed DRA also alleviates the saddle-point issue effectively, indicating that DRA obtains more robust models.

### 2.2 Data augmentation and information transfer for long-tailed data

Data augmentation (DA) on balanced datasets (Cubuk et al., 2020; Yun et al., 2019), such as Mixup (Zhang et al., 2017) and ISDA (Wang et al., 2019), have been adopted to improve long-tailed learning (Ahn et al., 2023; Xu et al., 2021; Zhong et al., 2021; Li et al., 2021). Moreover, images (Zada et al., 2022) or features (Li et al., 2022a) perturbed by noise and out-of-distribution data (Wei et al., 2022) could surprisingly serve as augmentation(FA) for imbalanced data, though they are harmful to balanced training. Head-to-tail transfer are highly explored recently (Yin et al., 2019; Liu et al., 2021; 2020; Kim et al., 2020). GIT (Zhou et al., 2022) uses GAN (Goodfellow et al., 2014) to learn nuisance transformations to transfer invariance for tail classes while CMO (Park et al., 2022) randomly pastes image patches of head classes on tail classes to transfer context information. Recently Ahn et al. (2023) and Hong et al. (2022) claim that class-wise or instance-aware augmentation is more effective, which is consistent with our findings in Section 3. The relation and difference between related data augmentation works and our proposed method will be discussed later.

## 2.3 Ensemble methods for long-tailed data

Ensemble methods for long-tailed recognition are based on multiple experts to take full advantage of limited data. Zhou et al. (2020) gradually fuse two branches for balanced and re-balanced label distributions respectively by a cumulative learning strategy while Cai et al. (2021) tend to make experts cope with specific and overlapping class splits to complements for each other. RIDE(Wang et al., 2020) proposes a strategy to route experts dynamically, and explicitly encourages the diversity of them. SADE(Zhang et al., 2022) allows each expert to adapt to various label-distribution and aggregates them by the consistency of their predictions in the test time, and could generalize to various test label distributions. In this work, we see ensemble methods as orthogonal to our proposed DRA, as introducing distributional robustness to ensemble methods is complex and under-explored even for balanced datasets. Exploring how much ensemble methods adapt to CCD shift and enhancing them from CCD shift view are left for future works.

## 2.4 Distributionally Robust Optimization (DRO)

DRO (Kuhn et al., 2019; Delage & Ye, 2010; Shafieezadeh-Abadeh et al., 2019) aims at generalization under distribution shift. It focuses on worst-case risk minimization across a set of potential distributions, known as uncertainty sets, determined based on various computations, e.g. Wasserstein distance (Kuhn et al., 2019), F-divergence (Namkoong & Duchi, 2016) and moments constraints (Delage & Ye, 2010). WRM (Sinha et al., 2017) gives a general solution to the DRO problem with Wasserstein distance uncertainty set. It converts the Lagrange penalized problem to a min-max optimization with a constant Lagrange multiplier (Algorithm 2 in Appendix). Lin et al. (2022) surveys recent studies on DRO. Our DRA generalizes WRM by using class-wise uncertainty set and generating a sequence of examples in the inner-optimization, which shows superiority over WRM theoretically and empirically. In the scope of long-tailed recognition, DRO-LT (Samuel & Chechik, 2021) utilizes DRO to alleviate the bias in feature learning which is closely related to our proposed DRA. However, it is significantly different from ours in the motivation, method, and adaptability. See a detailed discussion in Section 4.3.

# 3 Class-Conditional Distribution Shift in Long-Tailed Recognition

In this section, we conduct empirical studies on unreliable class-conditional distribution (CCD) estimation in long-tailed recognition. We first introduce problem setup. Then we proposed a new metric to measure the offset between empirical CCD and the ideal CCD. Finally, we present our findings and analysis of CCD shift issue based on the experimental results.

## 3.1 Problem setup

In classification/recognition setting, a labeled instance is a pair $(x, y)$, where $x \in \mathcal{X} \doteq \mathbb{R}^m$ and $y$ takes value in $[L] \doteq \{1, ..., L\}$ with $L$ the number of classes. We consider a $\theta$-parameterized classifier $f$, e.g. a neural network, which outputs $f_\theta(x) \in \mathbb{R}^L$ to estimate $P(y|x)$. Denote training set as $\{(x_i, y_i)\}_{i=1}^N \sim P_{train}$, with $P_{train} = \sum_{j \in [L]} P(y_j) P(x|y_j)$, $N_j$ the number of instances from class $j$, $P_{N,LT} = \sum_{j \in [L]} P(y_j) P_{N_j, LT}(x|y_j)$ the empirical distribution while $P_{N_j, LT}(x|y_j) = \frac{1}{N_j} \sum_{i: y_i = y_j} \delta(x_i, y_i)$ where $\delta$ is the Dirac-distribution. In the long-tailed scenario, $P(y)$ is imbalanced, and we assume that $P(y_1) > ... > P(y_L)$. In practice, the model is desired to perform well on all classes for e.g. detecting rare species or making fair decisions (Van Horn & Perona, 2017; Hinnefeld et al., 2018). Hence, the distribution for evaluation is usually class-balanced, i.e. $P_{test} = \frac{1}{L} \sum_{j \in [L]} P(x|y_j)$, In other words, the goal is minimizing the balanced risk (Menon et al., 2020)

$$R_{bal} = \frac{1}{L} \sum_{j \in [L]} P_{x|y_j}(y_j \neq \arg\max_{y \in [L]} \{f_\theta(x)\}_y) \tag{1}$$

$$= \frac{1}{L} \sum_{j \in [L]} E_{P(x|y_j)}[l_{0/1}(\arg\max_{y \in [L]} \{f_\theta(x)\}_y, y_j)].$$

### 3.2 Quantifying CCD shift

In previous work, to estimate $R_{bal}$, $P(x|y_j)$ is approximated by the empirical CCD (Menon et al., 2020; Vapnik, 1991) $P_{N_j,LT}(x|y_j)$, which is intuitively unreliable due to scarce instances of tail classes. The following proposition explains it more formally.

**Proposition 1.** *For a loss function $l(x,y)$ for any $y$ L-Lipschitz respect to $x$, $P(x|y)$ is a conditional distribution on $\mathbb{R}^m$ and $P_N(x|y)$ is the empirical distribution of $P(x|y)$ estimated from $N$ instances. Denote $A \doteq E_{P(x|y)}[\|x\|^\alpha] < \infty$ for some $\alpha > 0$, then $\exists\, c_1, c_2$, depending on $A$, $\alpha$, we have $\left| E_{P(x|y)}[l(x,y)] - E_{P_N(x|y)}[l(x,y)] \right| < t$ with probability at least $1 - c_1/e^{(\frac{t}{L})^{min\{m,a\}}c_2 N}$.*

It says with very few instances e.g. in tail classes, training performance does not ensure generalization, so the shift between empirical distribution and true distribution cannot be ignored. That is the so-called CCD shift. The proof of this proposition can be found in Appendix A.1. We also provide visualization and analysis of a toy example in Appendix B.1 to validate Proposition 1 and the consequence of CCD shift. Unfortunately, unreliability of CCD cannot be directly computed from above bound due to unavailable real distribution in practice.

Nevertheless, there is a way to quantify CCD shift: estimating CCD with more instances besides the long-tailed dataset as an oracle, which is more reliable by Proposition 1. Specifically, we choose CCD from the balanced dataset as the oracle, i.e. sampling from the following distribution

$$P_{remove\ shift}(x,y) = \sum_{j \in [L]} P_{LT}(y_j) P_{N_j,bal}(x|y_j), \tag{2}$$

named as *removing shift sampling.* Note that, by sampling the same number of instances as original long-tailed datasets from distribution (2), the CCD is changed while label distribution is kept still for a valid ablation. When training with removing shift sampling, more novel instances are seen by the model. One can use the accuracy gap between models trained with and without this sampling as a metric to measure the CCD shift between training and test. If the empirical CCD is reliable enough to estimate the true CCD, a more reliable CCD introduced by $P_{remove\ shift}$ would not bring significant improvement and the accuracy gap metric would be small. Conversely, a large gap, as on the long-tailed datasets in our experiments, means a significant CCD shift.

Specifically, we perform ablation experiments on CIFAR10/100-LT (imbalance ratio:100) without or with removing shift sampling to investigate the effect of CCD shift on *vanilla Empirical Risk Minimization* (ERM) (Vapnik, 1991) and other representatives of main-stream re-balancing methods including

- *Deferred Re-Weighting* (DRW) (Cao et al., 2019) uses class-balanced re-weight at the later stage of training i.e. after training an initial model by ERM, as a representative of deferred re-balance methods (Zhou et al., 2020; Cao et al., 2019; Park et al., 2021).

- *Classifier Re-Training* (CRT) (Kang et al., 2019) re-trains the linear classifier of an initial model trained by ERM with the feature extractor frozen, as a representative of decoupling methods (Kang et al., 2019; Zhang et al., 2021a; Zhong et al., 2021).

- *Post-Compensated Softmax* (PC softmax) (Hong et al., 2021) modifies the logits according to the training in the inference phase as $f_\theta^{PC}(x)_j = f_\theta(x)_j + log(P(y_j))$, serving as a special case of post-hoc logit adjustment (Menon et al., 2020) and a representative of logit adjustment methods (Hong et al., 2021; Menon et al., 2020; Li et al., 2022b).

We put more details of empirical study on removing shift sampling in Appendix B.

Table 1: Accuracy of decoupling method CRT on CIFAR10-LT under different settings without or with removing shift sampling. Underline means accuracy of imbalanced features and small font denotes difference between re-balanced and imbalanced features. More results are provided in Table 9 (in Appendix B).

| feature re-balance[1] | classifier adjust | base | removing shift |
|---|---|---|---|
| - | CRT | 77.72 | 87.07 |
| DRW | CRT | 75.97-1.75 | 88.42+1.35 |
| DRW | - | 78.06 | 89.01 |

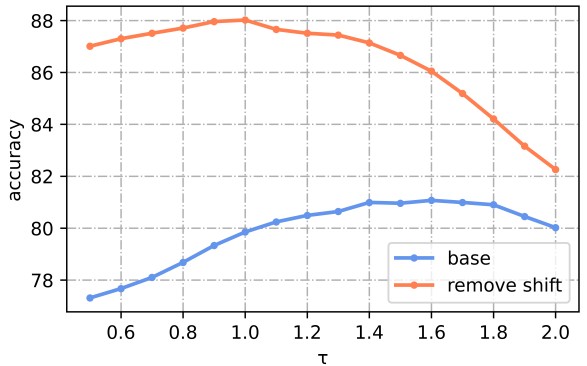

Figure 2: Accuracy on CIFAR10-LT with varying $\tau$ in post-hoc adjustment. After removing shift, $\tau = 1$ gets optimal performance while it is sub-optimal with shift of $P(x|y)$.

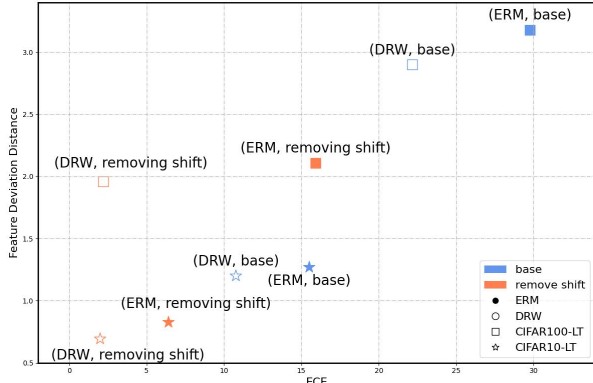

Figure 3: ECE and feature deviation distance with or without removing shift and re-balancing on CIFAR10/100-LT.

### 3.3 Empirical results and analysis

**CCD shift matters and is class-wise.** Figure 1 and Figure 9 (in Appendix B) quantitatively show how much CCD shift affects the performance in long-tailed recognition. The accuracy curves of all methods (vanilla ERM and re-balancing methods) exhibit significant gaps in performance with or without removing CCD shifts and the gap is much larger in classes with fewer instances. This confirms that CCD shift 1) is a key factor that limits the performance in long-tailed recognition; 2) is class-wise and more severe in tail classes. It motivates us to explicitly address the unreliable empirical distribution and consider varying reliability across classes.

**Decoupling methods benefits by avoiding more severe CCD shift.** A surprising phenomenon in decoupling methods is im-balanced features are more generalizable than re-balanced ones, as in Table 1. But it disappears once the CCD shift is removed, with re-balanced features leading to better performance than imbalanced ones. We explain that re-balancing in feature learning assigns higher weights to more unreliable tail classes thus exacerbating overall unreliablility of the empirical distribution i.e. decoupling overperforms simply re-balancing without introducing more CCD shift. While decoupling avoids aggravating CCD shift in feature learning, the shift remains and subsequent classifier adjustment may still suffer from it, leaving space to be improved as in Samuel & Chechik (2021); Zhang et al. (2021a); Wang et al. (2021).

**Sub-optimality of Fisher-consistent post-hoc adjustment is blamed to CCD shift.** Recall that post-hoc logit adjustment adjusts logits to $f_\theta(x)_j + \tau \log(P(y_j))$. The parameter $\tau = 1$ is proved Fisher-consistent, however, sub-optimal in experiments (Menon et al., 2020). We demystify this contradiction from the empirical study of CCD shift. Figure 2 shows logit adjustment gets best performance with $\tau$ much bigger than 1 and, however, with removing shift, $\tau = 1$ becomes optimal. We explain that rare instances from tail

---

[1]We choose DRW to get re-balanced features instead of RW (direct re-weighting) for RW gets the same result whether with removing shift sampling or not: it harms first-stage learning. The related explanation can be found in Appendix B.2.

classes are regarded as "out-of-distribution" examples and given low confidence to the true class (Wang et al., 2021). So more adjustment to tail classes, i.e. $\tau > 1$, adapts to hardly recognizable samples due to CCD shift. This explains the effectiveness of searching adjustment parameters in margin-based adjustment loss (Li et al., 2022b; Kini et al., 2021) as well. Moreover, our findings suggest that *adjustment and data augmentation shall be instance-level discriminative to compensate for samples hard to be recognized due to CCD shift*, agreeing with sample-aware augmentation of recent works (Hong et al., 2022).

**CCD shift and imbalance label distribution jointly affect confidence calibration and feature deviation.** We additionally study the influence of CCD shift on two issues that are found affected by long-tailed distribution by prior works, *confidence calibration* (Xu et al., 2021; Zhong et al., 2021) and *feature deviation* (Ye et al., 2021). Xu et al. (2021); Zhong et al. (2021) find that long-tailed distribution makes the calibration of models worse i.e. the output score inaccurate to estimate real likelihood, while Ye et al. (2021) observes that the distance of feature centers (average of all instances) between training and testing data is enlarged by imbalanced distribution, especially for classes with fewer instances. We use ECE (Expected Calibration Error) and feature deviation distance as metrics for these issues. See Appendix B.5 for introduction to these metrics and more discussion. From Figure 3, the main observation is that both confidence calibration and feature deviation are co-influenced by CCD shift and imbalanced label distribution. Utilizing removing shift sampling or re-balancing method (e.g. DRW) alone obtains much smaller ECE and feature deviation distance, while joint use of them gets even better results. These results imply that CCD shift is complementary to label imbalance, and addressing CCD shift is crucial for not only recognition but also other tasks requiring reliable feature representation.

# 4 Distributionally Robust Augmentation against CCD Shift

## 4.1 Formulating class-aware robustness via min-max optimization

Shift of CCD has been shown significantly limiting the generalization under long-tailed distribution even with existing re-balancing strategies. Though existing data augmentation could synthesize novel instances for tail classes, they do not target CCD shift directly, and thus may be still biased by the shift and cannot restore the real CCD convincingly (Hong et al., 2022; Ahn et al., 2023).

To this end, we explore addressing this problem from another perspective: **making models robust to CCD shift** (Some prior augmentation methods can also be understood in this scope, please refer to our discussion among those and ours in Sec 4.3). We utilize Distributional Robust Optimization (DRO) (Sinha et al., 2017; Delage & Ye, 2010) to obtain robustness against potential CCD shift, considering the worst risk among a set of potential distribution shift near the empirical distribution instead of trusting it totally (Menon et al., 2020; Xu et al., 2021; Cao et al., 2019). Formally, we aim at minimizing the DRO risk:

$$R_{DRO} = \frac{1}{L} \sum_{j \in [L]} \sup_{\hat{P}_j \in \mathcal{P}_j} E_{\hat{P}_j}[l_\theta(x, y)] \tag{3}$$

with $\hat{P}_j$ a probability measure on $\mathcal{X} \times [L]$, $l_\theta(x, y) = l(f_\theta(x), y)$, and $\mathcal{P}_j = \{\hat{P}_j | W_c(\hat{P}_j, P_{N,LT}(x|y_j)) < r_j\}$. $W_c$ is Wasserstein distance induced by cost function $c((x_1, y_1), (x_2, y_2)) = c_x(x_1, x_2) + \infty \cdot \mathbf{1}_{\{y_1 \neq y_2\}}$ where $c_x(x_1, x_2)$ is nonnegative, lower semi-continuous, and satisfies $c_x(x, x) = 0$ as a valid cost function to make the Wasserstein distance well-defined(Sinha et al., 2017; Villani, 2009). Additionally, we assume $c_x(x_1, x_2)$ is continuous and $c_x(\cdot, x_2)$ is 1-strongly convex for any $x_2$ and typically set $c_x(x_1, x_2) = \|x_1 - x_2\|_2^2$ as it meets the above properties to establish our following theoretical derivation. Different from previous methods (Sinha et al., 2017), **the radii of uncertainty sets $\mathcal{P}_j$ depend on classes to adapt to imbalanced instance numbers**, which implies different reliabilities by Proposition 1. Specifically, they are decreased with the increasing number of instances, i.e. $r_1 < \cdots < r_L$, to make the model robust enough without over-pessimism (Frogner et al., 2021).

Two critical questions about the DRO risk are: how to make it tractable and whether it is effective in solving CCD shift. For the former, we convert the intractable (3) on the set of potential distributions to a min-max optimization problem on the training set:

**Theorem 2** (Simplified version of Theorem 7). *For a loss function $l_\theta(x,y)$ continuous on $x$ for any $y$, our DRO risk equals*

$$R_{DRO} = \inf_{\lambda_j \geq 0, j \in [L]} \frac{1}{L}\Big\{ \sum_{j \in [L]} \lambda_j r_j + E_{P_{N,LT}(x|y_j)}\Big[\sup_{z=(x_z,y_z)} l_\theta(x_z, y_z) - \lambda_j \cdot c(z,(x,y))\Big]\Big\} \tag{4}$$

*with $z = (x_z, y_z) \in \mathcal{X} \times [L]$*

See the full Theorem 7 and proof in Appendix A.2. Motivated by previous DRO (Sinha et al., 2017), we relax $\lambda_j$ to hyperparameters and omit $r_j$ in (4) to simplify the optimization procedure. Then our objective changes from (4) to:

$$F(\theta) := \frac{1}{L} \sum_{j \in [L]} \sup_{\hat{P}_j}\{E_{\hat{P}_j}[l_\theta(x,y)] - \lambda_j \cdot W_c(\hat{P}_j, P_{N_{LT},j})\} \tag{5}$$

$$= \frac{1}{L} \sum_{j \in [L]} E_{P_{N,LT}(x|y_j)}\Big[\sup_{z=(x_z,y_z)}\{l_\theta(x_z, y_z) - \lambda_j \cdot c(z,(x,y))\}\Big] \tag{6}$$

The final objective (6) actually equals to the Lagrange penalty (5) for the original problem (3) as above. We will explain the optimization process in Section 4.2.

To show the rationality of the DRO risk, we will see that with mild conditions , the risk on balanced label distribution with real CCD is partially bounded by our converted objective (6):

**Theorem 3** (Simplified version of Theorem 14). *For any positive multipliers $\{\lambda_j\}_{j \in [L]}$, with mild conditions for the class-conditional distribution $P(x|y_j)$ of each class i.e. a certain order moment exists, with probability $1 - \eta$ the balanced risk (1) is bounded by*

$$R_{bal} \leq \frac{1}{L} \sum_{j \in [L]} \lambda_j C(N_j; \eta) + F(\theta), \tag{7}$$

*with $C(N_j; \eta)$ a constant only depending on $N_j$ and $\eta$.*

In fact, $C(N_j; \eta)$ is from the convergence rate between empirical CCD and real CCD(Fournier & Guillin, 2015), indicating the unreliability of empirical distribution of each class. It decreases as $N_j$ increases.

The bound shows that CCD shift is essential for long-tailed distribution and our method provides a principled solution to the CCD shift. It also reflects our insight of class-aware radius, since more robustness for tail class $j$ by scaling up $\lambda_j$ would reduce more risk as the factor $C(N_j)$ is larger. We put the full theorem, its proof and related remarks of the above bound in Appendix A.4.

## 4.2 Learning with a sequence of augmented data

The min-max optimization (6) can be solved by two bilevel steps: solving the inner max-optimization and computing the outer loss by its optimal solution:

$$\min_\theta \frac{1}{L} \sum_{j \in [L]} E_{P_{N,LT}(x|y_j)}[l_\theta(x^{aug}, y^{aug})] \tag{8}$$

$$s.t. \ (x^{aug}, y^{aug}) = \arg\max_{z=(x_z,y_z)}\{l_\theta(x_z, y_z) - \lambda_y \cdot c(z,(x,y))\}. \tag{9}$$

Specifically, we can apply gradient descent and obtain the last example in the inner optimization loop as $(x^{aug}, y^{aug})$[1]. This process can be understood as data augmentation adaptive to current model and training instance, as indicated by the objective: $l_\theta(x_z, y_z)$ can be seen as mining harder examples for the current

---

[1]In fact, with our cost $c$, $y^{aug} = y$ for each $(x,y)$ in (9), (11), as $y^{aug} \neq y$ turns objective of inner optimization to negative infinity.

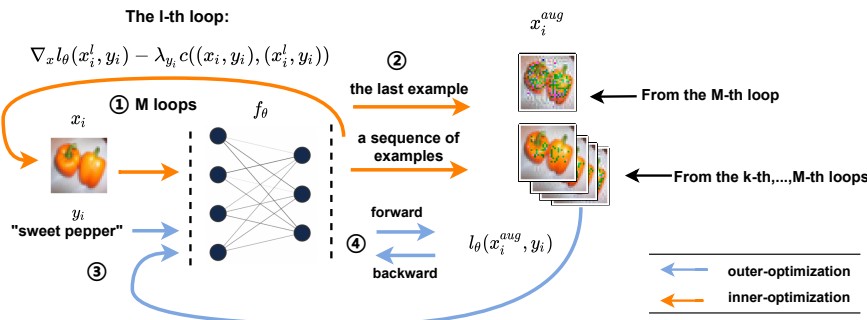

Figure 4: An illustration of DRA. Augmentation examples $x_i^{aug}$ generated from original data $x_i$ in $M$ inner loops (with two choices of the last example or a sequence of examples) are used as new training instances $(x_i^{aug}, y_i)$ to train a robust model against potential distribution shift. The order numbers ①-④ indicate the process of DRA in an overall iteration.

model, with $c(z, (x, y))$ constraining examples not too far away from the original instance and $\lambda_j$ determining the magnitude of the constraint.

However, as found in our experiments (shown in Figure 5), this strategy causes unstable optimization and bad performance when $\lambda_j$ is relatively small, since it brings smaller convergence rate in the inner optimization thus the gradient computed for the last point is biased for outer optimization. See a more detailed discussion on this result in Remark 2 in Appendix A.3. To overcome these limitations, **we propose DRA Algorithm** (as illustrated Figure 4), **which uses a sequence of examples, i.e. the last $s := M - k$ points, from the loops of inner-optimization**, annotated as "a sequence of examples" in Figure 4. The overall optimization becomes:

$$\min_\theta \frac{1}{L} \sum_{j \in [L]} E_{P_{N,LT}(x|y_j)} \Big[ \sum_{l \in [s]} \frac{1}{s} l_\theta(x^{aug,l}, y^{aug}) \Big] \tag{10}$$

$$s.t. \{(x^{aug,l}, y^{aug})\}_{l \in [s]} \text{ from the last } s \text{ loops in } \max_{z=(x_z,y_z)} \{l_\theta(x_z, y_z) - \lambda_y \cdot c(z, (x, y))\}. \tag{11}$$

Figure 4 and Algorithm 1 (in Appendix C.2) explain the inner optimization in detail by illustration and pseudo-code respectively.

Empirically, our method achieves better performance with more stable optimization (as shown in Figure 5 and Table 5). Theoretically, we can prove a sequence of examples can make the optimization more stable i.e. as $s$ increases, the following convergence bound becomes more tight.

**Theorem 4** (Simplified version of Theorem 9). *Under common conditions from WRM(Sinha et al., 2017), there exist constants $C_1 < 1, C_2$ making below convergence bound valid:*

$$\frac{1}{T} \sum_{t \in [t]} E[\|\nabla_\theta F(\theta_t)\|_2^2] < \frac{1 - (C_1)^s}{sN_{batch}} C_2 \epsilon + O\Big(\sqrt{\frac{1}{N_{batch}T}}\Big), \tag{12}$$

*with $N_{batch}$ the batch size used and $\epsilon$ is the step tolerance of inner optimization as in Algorithm 1.*

The bound is formally presented in Theorem 9 in Appendix A.3, where more theoretical analysis and comments are provided.

### 4.3 Discussion with prior works

**Relation to other augmentation methods.** As clarified before, DRA pursues robustness to CCD shift by generating novel examples as data augmentation. Some prior augmentation works can be understood as improving robustness to a family of potential distributions, e.g. GIT(Zhou et al., 2022) can be seen

as increasing robustness to the family of transformed (by e.g. luminance changes and rotations) empirical distributions while Open sampling (Wei et al., 2022) introduces additional open-set data, bringing robustness to potential unseen subpopulation shift (Duchi et al., 2022). However, the enhanced robustness relies on specific handcrafted operations or auxiliary data e.g. transformations in GIT and open-set datasets in Open Sampling, which may incur additional bias and cannot address CCD shift. Our DRA is a more principled method considering robustness under a ball of distribution discrepancy (Wasserstein distance (Sinha et al., 2017)), which is theoretically substantiated to handle CCD shift without additional modules and data.

**Relation to DRO-LT.** Prior work DRO-LT (Samuel & Chechik, 2021) also utilizes DRO to improve long-tailed recognition against instance scarcity. Here, we summarize the main differences between DRO-LT and our proposed DRA. *1) Motivation*: our work offers a comprehensive analysis of CCD shift from instance scarcity on existing long-tailed solutions, while DRO-LT aims at remedying biased representation of tail classes in decoupling methods (Kang et al., 2019), which is a derived issue of CCD shift, as found in our study (Sec 3.3). *2) Method*: DRO-LT considers a feature-level uncertainty set of Gaussians which is oversimplified to cover the real test distributions and thus fails to give a theoretical analysis between enhanced robustness and generalization. DRA considers uncertainty sets under Wassertein distance, which admits a novel generalization bound and preserves the advantage of DRO under Wasserstein distance, i.e. customized robustness by special cost function (Villani, 2009; Sinha et al., 2017). Additionally, DRO-LT estimates its objective by a margin-based loss, which slows convergence similar to other margin-based loss (Ye et al., 2020; Li et al., 2022b), while the convergence of DRA does not require additional training phase and epochs. *3) Adaptability*: DRA can easily integrate with various prior methods while DRO-LT is coupled with centroid classifier and classifier re-training strategy. The latter may not be applicable to other tasks, e.g. object detection (Zhang et al., 2021a), and centroid classifier is rarely used compared with softmax classifier.

# 5 Experiments

Experiments on long-tailed classification tasks show that DRA significantly improves existing re-balancing strategies, even on strong baselines trained with mixup and RandAugment. More thorough analysis shows the effectiveness of DRA on addressing confidence calibration (Guo et al., 2017) and saddle-point issue (Rangwani et al., 2022b). Complete experimental details and more results including visualization of examples from DRA are provided in Appendix C.

## 5.1 Experimental setup

**Datasets.** We conduct experiments on CIFAR10-LT, CIFAR100-LT, Tiny-ImageNet-LT (Cao et al., 2019) CelebA-5 (Kim et al., 2020) and ImageNet-LT (Liu et al., 2019). The imbalance ratio of CIFAR-LT and Tiny-ImageNet-LT is set to 100. As in prior studies (Cao et al., 2019; Wei et al., 2022), we report the Top-1 accuracy on the test set for Celeb-5 and on the validation set for other benchmarks. Results of ImageNet-LT are reported on three splits of classes: Many-shot (more than 100), Medium-shot (20-100) and Few-shot (less than 20). For specific experimental settings for various datasets, please see Appendix C.1.

**Comparative methods.** We compare our methods with four groups of related methods: (I) re-balancing methods including Decoupling (Kang et al., 2019), Logit Adjustment (Menon et al., 2020), PC softmax (Hong et al., 2021), DRS/DRW (Cao et al., 2019), and (II) their variants: LADE (Hong et al., 2021), IB (Park et al., 2021), Vector loss (Kini et al., 2021), Auto-balance (Li et al., 2022b), CDT (Ye et al., 2020), MFW (Ye et al., 2021), SAM (Rangwani et al., 2022b), CC-SAM(Zhou et al., 2023); (III) data/feature augmentation methods(DA/FA) including Mislas (Zhong et al., 2021), CUDA (Ahn et al., 2023), Open-sampling (Wei et al., 2022), DRO-LT (Samuel & Chechik, 2021), GCL(Li et al., 2022a); (IV) head-to-tail transfer methods including M2m (Kim et al., 2020), GIT (Zhou et al., 2022), SAFA (Hong et al., 2022), CMO (Park et al., 2022).

**Implementation of DRA.** Inspired by (Kim et al., 2020; Samuel & Chechik, 2021), we apply DRA in the later stage of training since we need an initial model that has fit the data distribution for DRA to generate

Table 2: Comparisons of accuracy (%) with Mixup and RandAugment on different benchmarks. Balanced Softmax is noted as BS. / means results with Mixup/RandAugment respectively. ∗: Mixup-based augmentation degrades Balanced Softmax on ImageNet-LT so DRA is not used with the union of them. ‡: we align our training epochs (i.e. 100/300) with CUDA/DRO-LT respectively for fair comparison. † means results from the original papers, others are reproduced by us. The best results are in bold.

| Method (Mixup / RandAug) | CIFAR10-LT | CIFAR100-LT | ImageNet-LT | | | |
|---|---|---|---|---|---|---|
| | | | Many | Med | Few | All |
| CE-DRS | 79.7 / 80.89 | 47.08 / 46.95 | 61.76 / 61.13 | 49.69 / 49.61 | 27.14 / 26.02 | 50.11 / 50.13 |
| CE-DRW-SAM† | 80.6 | 44.6 | 56.6 | 45.8 | 28.1 | 47.1 |
| CE-DRW-CUDA† | 80.54 | 46.76 | 61.8 | 48.3 | 30.3 | 51.0 |
| CE-DRW-CMO | 81.98 | 47.05 | 60.8 | 48.6 | 35.5 | 51.2† |
| SAFA† | 80.48 | 46.04 | 63.8 | 49.9 | 33.4 | 53.1 |
| CE-DRS + Ours | 80.59 / 82.59 | 47.39 / 47.68 | 61.77 / 61.30 | 50.23 / 49.73 | 27.66 / 26.14 | 50.89 / 50.42 |
| CE-DRW + CUDA + Ours | 81.83 | 47.71 | 61.93 | 49.45 | 31.11 | 51.74 |
| BS | 81.23 / 81.22 | 46.63 / 46.81 | 63.0 / 63.16 | 49.80 / 50.85 | 27.62 / 30.35 | 51.5* / 52.27‡ |
| BS-CutMix | 81.57 | 46.46 | - | - | - | - |
| BS-CUDA | 81.35 | 46.89 | 61.9 | 49.2 | 32.3 | 51.6 |
| BS-CMO | 82.30 | 46.72 | 62.0 | 49.1 | **36.7** | 52.3† |
| BS + Ours | 82.21 / 81.83 | 48.11 / 47.69 | - / 63.32 | - / 51.22 | - / 30.87 | - / 52.77 |
| BS + CUDA + Ours | 82.50 | 48.71 | 62.23 | 49.84 | 33.07 | 52.32‡ |
| DRO-LT† | 82.6 | 47.31 | 64.0 | 49.8 | 33.1 | 53.5 |
| BS (300 epochs)‡ | - / 82.19 | - / 47.97 | - / 63.83 | - / 51.53 | - / 32.08 | - / 53.61 |
| BS + Ours (300 epochs)‡ | - / **83.41** | - / **48.92** | - / 63.89 | - / **52.38** | - / 34.46 | - / **54.30** |
| PC Softmax | 81.29 / 80.82 | 46.12 / 46.20 | 62.78 / 63.09 | 50.37 / 51.15 | 30.57 / 30.36 | 52.25 / 52.41 |
| PC Softmax + Ours | 82.41 / 81.41 | 46.35 / 46.77 | 63.09 / 62.68 | 51.36 / 51.89 | 32.20 / 31.58 | 53.27 / 52.89 |
| CRT | - | - | 62.77 / 64.11 | 49.51 / 50.77 | 31.75 / 28.46 | 51.77 / 52.36 |
| CRT-CUDA† | - | - | 62.3 | 47.2 | 28.4 | 50.2 |
| MISLAS | 82.1 | 47.21 | 63.3 | 51.21 | 33.73 | 53.16 |
| CC-SAM† | 82.41 | 48.77 | 61.4 | 49.5 | 37.1 | 52.4 |
| GCL | 81.57 | 47.52 | - | - | - | 54.12† |
| CRT + Ours | - | - | **64.52** / 64.24 | 50.87 / 50.71 | 33.01 / 31.84 | 53.65 / 52.89 |

augmentation examples. For the class-wise multiplier $\lambda_{j\, j\in[L]}$, we set $\lambda_j \propto N_j{}^\beta * S$ with $\beta, S \geq 0$ where $\beta$ determines robustness difference over classes and $S$ determines overall robustness level, which reduces the multipliers of DRA to only two hyperparameters, regardless of the number of classes considered. $M$, $\alpha_{inner}$ in DRA (as in Algorithm 1) are set by our default value and not considered as hyperparameters for tuning while $k$ is set as $M/2$ or $M-1$ corresponding to a sequence or a single example respectively. Full implementation details of DRA are in Appendix C.2.

## 5.2 Main results

**DRA cooperating with re-balancing strategies.** As summarized in Table 3, 4, and 12 (in Appendix C.3), DRA improves re-balancing baselines significantly on the five benchmarks, building stronger baselines for them. The improvement on CIFAR-LT is more significant under the setting of prior work (Zhou et al., 2022) with smaller batch size, e.g. overperforming previous state-of-the-art LADE (See Table 14 in the Appendix C.4). These strongly supported that the distributional robustness introduced by our DRA benefits generalization under CCD shift. Moreover, the gains by DRA on ImageNet-LT is not only from few-shot but also from many-shot and medium-shot, verifying distributionally robustness helps the generalization of head classes as well.

**DRA cooperating with existing data augmentation.** Recently, solutions for long-tail learning often rely on augmentation schemes from balanced datasets (Samuel & Chechik, 2021; Park et al., 2022; Li et al., 2022a). This makes it unclear whether the improvement from them comes from alleviating CCD shift or, better regularization inherent and induced bias of relied data augmentation. To accurately evaluate DRA and make a fair comparison, we examine DRA's performance in conjunction with two data augmentation from balanced learning, Mixup (Zhang et al., 2017) and RandAugment (Cubuk et al., 2020), and a recent strong data augmentation solution designed for long-tailed recognition, CUDA(Ahn et al., 2023). As in Table 2, DRA even brings significant improvement for these strong baselines trained with other data augmentations on CIFAR-LT and ImageNet-LT, achieving superior or comparable performance to state-of-the-art data

Table 3: Comparison of results on CelebA-5.

| Method | Acc |
|---|---|
| ERM | 72.7 |
| LDAM-DRW | 74.5 |
| M2m-LDAM | 75.6 |
| Open Sampling | 78.6 |
| CE-DRS | 73.1 |
| +Ours | 75.2 |
| LDAM-DRS | 75.9 |
| +Ours | **80.1** |
| PC softmax | 76.2 |
| +Ours | 77.1 |
| Balanced Softmax | 76.8 |
| +Ours | 77.6 |

Table 4: Comparison of accuracy (%) on different benchmarks. Balanced Softmax is noted as BS. The best results are in bold and small red font denotes performance gain.

| Method | CIFAR10-LT | CIFAR100-LT | Method | ImageNet-LT | | | |
|---|---|---|---|---|---|---|---|
| | | | | Many | Med | Few | All |
| ERM | 73.24 | 40.28 | ERM | **66.84** | 40.89 | 11.54 | 46.05 |
| PC softmax | 79.58 | 44.34 | PC softmax | 62.13 | 49.25 | 30.51 | 51.18 |
| +Ours | **80.39**+0.81 | 44.84+0.50 | +Ours | 62.41 | 49.72 | 31.89 | 51.64+0.46 |
| BS | 79.65 | 45.17 | BS | 62.70 | 48.81 | 31.62 | 51.82 |
| +Ours | 80.33+0.68 | **45.71**+0.54 | +Ours | 63.09 | **49.58** | **32.17** | **52.42**+0.60 |
| CE-DRS | 77.21 | 44.16 | CRT | 61.59 | 47.70 | 30.32 | 50.3 |
| +Ours | 78.89+1.68 | 44.87+0.71 | +Ours | 61.81 | 49.43 | 31.57 | 51.37+1.07 |
| LDAM-DRS | 78.71 | 45.12 | - | - | - | - | - |
| +Ours | 79.75+1.04 | 45.54+0.42 | - | - | - | - | - |

augmentation or head-to-tail transfer methods. Note that DRA integrates with these data augmentation without modifications but surpasses existing solutions based on them, e.g. Mislas and CMO. This reflects that existing methods are still biased by CCD shift even with modifications for adapting to long-tailed recognition.

### 5.3 Ablation study

**Class-aware uncertainty set.** In the implementation of DRA, class-aware multiplier are set as $\lambda_j \propto N_j{}^\beta * S$ with hyperparameters $\beta$ and $S$. Through $\beta$ various robustness are assigned while $\beta = 0$ gives every class the same robustness, reducing DRA to existing WRM method(Sinha et al., 2017). As shown in Figure 5 and Figure 6, with different values of $\beta$ and $S$, we can reveal the effects of class-aware radius and validate effectiveness of ours. In Figure 5, class-aware radius exhibits significant superiority under multiple choices of $S$ while class-consistent radius (in WRM) only slightly improves the performance. Moreover, Figure 6 indicates class-aware radius ($\beta > 0$) brings significant improvement over class-consistent radius ($\beta=0$) and $\beta$ need to be adjusted carefully to ensure higher robustness for CCD shift.

**The number of augmentation examples.** To verify the effect of using a sequence of points as augmentation examples in DRA to calculate outer gradient,we set $k$ in DRA to $M/2$(a sequence) or $M-1$(single example) to obtain comparative results. Figure 5 shows the effectiveness of DRA with different numbers of augmentation examples. As the bound (12) indicates, a sequence of augmentation examples is more stable than the last point with relatively small $S$. Moreover, we probe if the effectiveness of our strategy is from using a sequence or examples not the last one. Specially, we compare a variant of randomly selecting an example within the last M/2 iteration, noted as 'random example' in Figure 5. The result of this variant is worse than a sequence of examples most of the time but overperforms the last example under smaller S, supporting our insight reflected by (12) that only using the last example is not stable for optimization.

**Other cost functions** Other than our typical cost function $c(\cdot, \cdot) = c_x(x_1, x_2) + \infty \cdot \mathbf{1}\{y_1 \neq y_2\}$ with $c_x(x_1, x_2) = \|x_1 - x_2\|_2^2$, we conduct experiments on other $c_x \in \{\|x_1 - x_2\|_p^r | p = 1, 2, 4, r = 1, 2\}$ as in prior work(Sinha et al., 2017). However, they are not 1-strongly convex so fails to apply our convergence guarantee (12) or the guarantee from WRM (Sinha et al., 2017). For these cost functions, as in Table 5, we indeed observe that the inner optimization becomes unstable and may cause lower performance than baseline or even diverge. Fortunately, a sequence of examples(k=M/2), as heuristics in these cases, also effectively alleviates this issue and gains consistent improvement, serving as appropriate augments for only the last example(k=M-1), indicating our strategy generalizing to more cost functions that prior work(Sinha et al., 2017) could not.

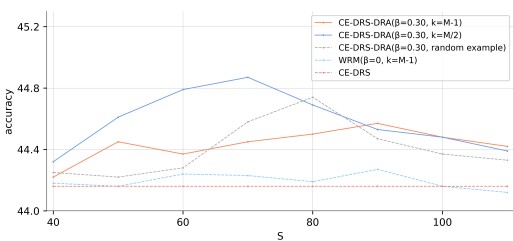
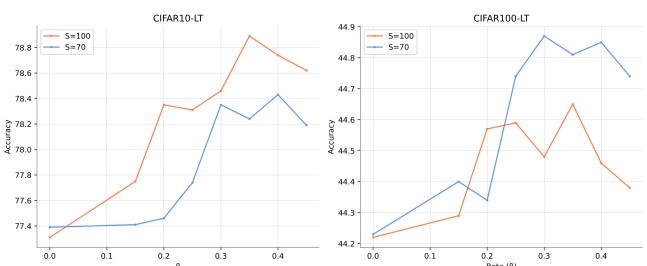

Figure 5: Accuracy on CIFAR100-LT with respect to different choices of hyperparameters $S$ and $k$ under $\beta = 0$ or $\beta = 0.30$. Random example means randomly selecting one from the last $M/2$ examples.

Figure 6: Accuracy of DRA on CIFAR10/100-LT with respect to different choices of $\beta$ under $S = 70$ or $S = 100$.

Table 5: Accuracy on CIFAR10/100-LT with respect to different choices of cost functions in the form of $c_x(x_1, x-2) = \|x_1 - x_2\|_p^r$ while $c((x_1, y_1), (x_2, y_2)) = c_x(x_1, x_2) + \infty \cdot \mathbf{1}_{\{y_1 \neq y_2\}}$. The baseline is CE-DRS and 'nan' means the training process does not converge.

| (r, k) | CIFAR10-LT | | | CIFAR100-LT | | |
|---|---|---|---|---|---|---|
| | p=1 | 2 | 4 | p=1 | 2 | 4 |
| (2, M-1) | nan | 78.89 | 77.78 | nan | 44.57 | 44.45 |
| (2, M/2) | nan | 77.83 | 78.56 | nan | 44.86 | 44.31 |
| (1, M-1) | 56.78 | 73.24 | 77.34 | 40.57 | 44.64 | 44.72 |
| (1, M/2) | 79.12 | 78.07 | 77.41 | 44.81 | 45.12 | 44.75 |

Table 6: ECE (%) on ImageNet-LT. Balanced Softmax is noted as BS. Smaller ECE means better calibration.

| Method | w/o mixup | | w/ mixup | |
|---|---|---|---|---|
| | - | +DRA | - | +DRA |
| CRT | 8.78 | 4.91 | 5.90 | 2.79 |
| PC softmax | 6.81 | 4.40 | 4.21 | 3.88 |
| BS | 6.29 | 3.04 | - | - |

### 5.4 More thorough study

**DRA on confidence calibration.** Inspired by our discovery that CCD shift degrades confidence calibration, we evaluate DRA's impact on confidence calibration with the Expected Calibration Error (ECE) metric (Guo et al., 2017). As in Table 6 and Figure 13, 14 (Appendix C.5), it is kind of surprising that robustness to CCD shift from DRA not only improves calibration but also improves the well-calibrated models with Mixup most of the time on the ImageNet-LT and CIFAR-LT (See a complete discussion in Appendix C.5).

**Alleviating the saddle-point issue.** To further prove that DRA would obtain solutions more robust to CCD shift, we analyze the loss landscape of solutions from DRA. Recent work(Rangwani et al., 2022b) analyzes the loss landscape of long-tailed learning on class-wise loss and finds that solutions converge to a saddle point in the landscapes of tail classes,

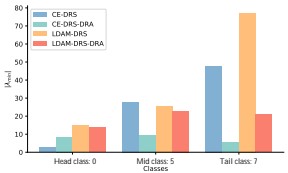
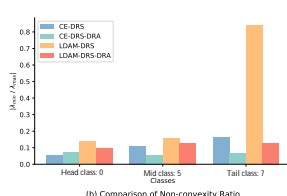

Figure 7: Comparison on $\lambda_{min}$ and $|\frac{\lambda_{min}}{\lambda_{max}}|$. DRA-trained models avoid convergence to non-convex region, as indicated by lower $\lambda_{min}$ and $|\frac{\lambda_{min}}{\lambda_{max}}|$.

which demonstrates poor generalization and instability to distribution shift(Dauphin et al., 2014). We evaluate DRA by the metrics proposed by Rangwani et al. (2022b), minimum negative eigenvalue of Hessian $|\lambda_{min}|$ and non-convexity ratio $|\frac{\lambda_{min}}{\lambda_{max}}|$ on CIFAR10-LT. As in Figure 7, DRA obtains solutions with significantly smaller $\lambda_{min}$ and $|\frac{\lambda_{min}}{\lambda_{max}}|$, which means that DRA finds more flat solutions and quantitatively prove that it can alleviate saddle point issue and obtain more robust models.

Table 7: Training cost comparison among DRA and previous methods on CIFAR100-LT. All values are in seconds. For two-stage methods, time per epoch and epochs for each stage are reported. Additional Time means extra time for pre-training modules beyond end-to-end training. The ratio of training time to baseline (CE-DRS) is also reported in overall time.

| Method | Time per epoch | Additional Time | epochs | Overall Time |
|---|---|---|---|---|
| ERM/CE-DRS | 2.85 | - | 200 | 637.0 (1.0×) |
| GIT | 2.85/15.34 | 32891.2 | 160+40 | 34010.2 (53.4×) |
| M2m | 2.85/24.75 | 637.0 | 160+40 | 2109.0 (3.3×) |
| CUDA | 5.26 | - | 200 | 1083.0 (1.7×) |
| RIDE(3 experts) | 27.59/6.11 | - | 200+5 | 5681.0 (8.92×) |
| DRO-LT | 2.85/62.3 | - | 200+100 | 6867 (10.78×) |
| SAM | 4.95 | - | 200 | 1044.0 (1.64×) |
| CC-SAM | 18.46 + 4.85 | - | 200+30 | 3837.5 (6.02×) |
| CE-DRS-DRA(k=M-1) | 2.85/19.37 | - | 160+40 | 1280.6 (2.01×) |
| CE-DRS-DRA(k=M/2) | 2.85/26.45 | - | 160+40 | 1543.8 (2.42×) |

**Training efficiency.** We make a comprehensive comparison of training cost among DRA and related previous methods on CIFAR100-LT in Table 7. All the experiments are conducted on an Nvidia RTX 2080Ti GPU. From the results, we find that 1) **long-tailed solutions using DA/FA/multi-expert more or less increase the cost of training**, e.g. CUDA and DRO-LT require additional time to compute augmentation score or class prototype each epoch respectively while our DRA introduces an extra inner optimization. 2) **The additional training cost of DRA is mild** as DRA is only equipped in the later stage of training (e.g. the last 40 epochs on CIFAR100-LT) and enjoys our theoretical convergence guarantee (i.e. no need for more epochs for convergence like DRO-LT), the overall cost is about only 2× CE-DRS/ERM, which is not a big burden, comparable to recent works CUDA (1.7×) and SAM (1.64×), and much less than existing solutions e.g. DRO-LT (10.78×) and GIT (53.4×). For other datasets, as the usage ratio of DRA is lower e.g. 30 epochs in the whole 180 epochs on ImageNet-LT, the extra training cost would be lower.

## 6 Conclusion

In this study, we present empirical evidence suggesting that unreliable distribution estimation in long-tailed recognition, exhibiting as shift of $P(x|y)$, is a key factor to limit performance even with re-balancing methods. Our findings indicate that only regarding long-tailed learning as label distribution shift problem is not sufficiently comprehensive and reviewed existing methods from the view of CCD shift. Through our proposed DRA, we show training a model robust to potential distribution shift can effectively improve long-tailed recognition, aligning with theoretical guarantees. Moreover, we highlight the potential of extending our concepts to the recently proposed multi-domain long-tailed settings(Tang et al., 2022; Gu et al., 2022), as appropriate distribution robustness can simultaneously improve instance scarcity and domain shift issues, which is promising to explore for future works. There is some space to improve DRA, e.g. more precise uncertainty set for potential shift or more efficient solutions to solve bi-level optimization. Overall, to some degree, our work sheds light on the bottleneck of long-tailed recognition and encourages the exploration of further solutions addressing the issue of unreliable empirical estimation.

## Acknowledgments

The authors would like to thank Ruibing Hou and Nan Kang for feedback on various drafts and helpful discussions. This work is partially supported by National Key R&D Program of China no. 2021ZD0111901, and National Natural Science Foundation of China (NSFC): 62376259.

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

# A Proofs and More Theoretical Analysis

**Lemma 5** (Kantorovich-Rubinstein Theorem). *((Villani, 2009), 5.16) For Wassernstein distance $W_c$ between two distributions $Q$ and $Q'$, $Lip(f) = \sup_{x_1,x_2} \frac{|f(x_1)-f(x_2)|}{\|x_1-x_2\|}$, it admits*

$$W_c(Q,Q') = \sup_{Lip(\phi)<1} \int_{\mathbb{R}^m} \phi(\xi)Q(d\xi) - \int_{\mathbb{R}^m} \phi(\xi)Q'(d\xi).$$

**Lemma 6** (Measure concentration). *((Fournier & Guillin, 2015), Theorem 2) For a probability distribution $P$ on $\mathbb{R}^m$ and $P_N$ is the empirical distribution of $N$ instances i.i.d sampled from $P$, $c(\cdot,\cdot)$ is a cost function inducing the Wasserstein Distance $W_c(c(x_1,x_2) := \|x_1 - x_2\|_p^r$ with $p \geq 1$, $r \in \{1,2\}$ is a special case), $A = E_{P(x|y)}\{c(x,x_0)^\alpha\} < \infty$ for some $\alpha > 0$ and $x_0 \in \mathbb{R}^m$, then $\exists c_1, c_2$, depending on $A$, $\alpha$, $P(W_c(P,P_N) < \epsilon) > 1 - \eta$ holds, for*

$$\epsilon(\eta) = \begin{cases} (\frac{log(c_1/\eta)}{c_2 N})^{1/m} & if\ N \geq \frac{log(c_1/\eta)}{c_2} \\ (\frac{log(c_1/\eta)}{c_2 N})^{1/\alpha} & if\ N < \frac{log(c_1/\eta)}{c_2} \end{cases}.$$

## A.1 Proof of Proposition 1

*Proof.*

$$\begin{aligned}
&\left| E_{P(x|y)}\{l(x,y)\} - E_{P_N(x|y)}\{l(x,y)\} \right| \\
=&L \cdot \left| E_{P(x|y)}\{l(x,y)/L\} - E_{P_N(x|y)}\{l(x,y)/L\} \right| \\
\leq&L \cdot W_c(P(x|y), P_N(x|y))
\end{aligned}$$

The last equality is from Lemma 5. And directly substituting $\eta$ in Lemma 6 by $\frac{c_1}{e^{(\frac{t}{L})^{min\{m,a\}}c_2 N}}$ gets the result. $\square$

### A.2 Theorem 7 and proof

**Theorem 7.** *For a loss $l_\theta(x, y) : \Theta \times (\mathcal{X} \times [L]) \to \mathbb{R}$ continuous with respect to $x$ for any $y$ and a valid cost function $c$, we have*

$$R_{DRO} = \inf_{\lambda_j \geq 0, j \in [L]} \frac{1}{L} \{\lambda_j r_j + \sum_{j \in [L]} E_{P_{N,LT}(x|y_j)} \{ \sup_{z=(x_z, y_z) \in \mathcal{X} \times [L]} l_\theta(x_z, y_z) - \lambda_j \cdot c(z, (x, y))\}\}. \quad (13)$$

*Considering Lagrange penalty with non-negative multipliers $\{\lambda_j\}_{j \in [L]}$, we have the reformulation:*

$$F(\theta) := \frac{1}{L} \sum_{j \in [L]} \sup_{\hat{P}_j} E_{\hat{P}_j} \{l_\theta(x, y)\} - \lambda_j \cdot W_c(\hat{P}_j, P_{N_{LT,j}}) \quad (14)$$

$$= \frac{1}{L} \sum_{j \in [L]} E_{P_{N,LT}(x|y_j)} \{ \sup_{z=(x_z, y_z) \in \mathcal{X} \times [L]} l_\theta(x_z, y_z) - \lambda_j \cdot c(z, (x, y))\}. \quad (15)$$

*Proof.* Our proof generalizes the proof in Sinha et al. (2017), which is a special case $\lambda_j \equiv C$ of ours. Using the duality firstly, we have

$$R_{DRO} = \frac{1}{L} \sum_{j \in [L]} \sup_{\hat{P}_j \in \mathcal{P}_j} E_{\hat{P}_j}[l_\theta(x, y)]$$

$$= \sup_{\hat{P}_j \in \mathcal{P}_j} \inf_{\lambda_j \geq 0} \frac{1}{L} \sum_{j \in [L]} \{E_{\hat{P}_j}[l_\theta(x, y)] - \lambda_j W_c(\hat{P}_j, P_{N,LT}(x|y_j)) + \lambda_j r_j\}$$

$$= \inf_{\lambda_j \geq 0} \{\lambda_j r_j + \sup_{\hat{P}_j \in \mathcal{P}_j} \frac{1}{L} \sum_{j \in [L]} \{E_{\hat{P}_j}[l_\theta(x, y)] - \lambda_j W_c(\hat{P}_j, P_{N,LT}(x|y_j))\}\}.$$

The last equality is based on strong duality which will be explained later. All we need to prove now is

$$\sup_{\hat{P}_j \in \mathcal{P}_j} \frac{1}{L} \sum_{j \in [L]} \{E_{\hat{P}_j}[l_\theta(x, y)] - \lambda_j W_c(\hat{P}_j, P_{N,LT}(x|y_j))\}$$

$$= \frac{1}{L} \sum_{j \in [L]} E_{P_{N,LT}(x|y_j)} [ \sup_{z=(x_z, y_z) \in \mathcal{X} \times [L]} l_\theta(x_z, y_z) - \lambda_j \cdot c(z, (x, y))].$$

Actually, we have

$$\sup_{\hat{P}_j \in \mathcal{P}_j} \frac{1}{L} \sum_{j \in [L]} \{E_{\hat{P}_j}[l_\theta(x, y)] - \lambda_j W_c(\hat{P}_j, P_{N,LT}(x|y_j))\}$$

$$= \sup_{\hat{P}_j \in \mathcal{P}_j} \frac{1}{L} \sum_{j \in [L]} \int l_\theta(\xi, y) \hat{P}_j(d\xi) - \lambda_j \inf_{\mathcal{M} \in \pi(\hat{P}_j, P_{N,LT}(x|y_j))} \int c(\xi_1, \xi_2) \, \mathcal{M}(d\xi_1, d\xi_2)$$

$$= \sup_{\hat{P}_j \in \mathcal{P}_j, \mathcal{M} \in \pi(\hat{P}_j, P_{N,LT}(x|y_j))} \frac{1}{L} \sum_{j \in [L]} \int l_\theta(\xi, y) - \lambda_j c((\xi, y), \xi_2) \, \mathcal{M}((d\xi, y), d\xi_2)$$

$$\leq \sup_{\hat{P}_j \in \mathcal{P}_j, \mathcal{M} \in \pi(\hat{P}_j, P_{N,LT}(x|y_j))} \frac{1}{L} \sum_{j \in [L]} \int \sup_{z=(x_z, y_z) \in \mathcal{X} \times [L]} l_\theta(x_z, y_z) - \lambda_j c((x_z, y_z), \xi_2) \, \mathcal{M}(d\xi_1, d\xi_2)$$

$$= \frac{1}{L} \sum_{j \in [L]} E_{P_{N,LT}(x|y_j)} [ \sup_{z=(x_z, y_z) \in \mathcal{X} \times [L]} l_\theta(x_z, y_z) - \lambda_j \cdot c(z, (x, y))].$$

The first equality is from the definition of Wasserstein distance and the last equality is from $\sup_{z=(x_z, y_z) \in \mathcal{X} \times [L]} l_\theta(x_z, y_z) - \lambda_j \cdot c(z, \xi_2)$ is independent of $\xi_1$. We now explain that the inequality is actually

an equality. For every $i$, we can find $z_i^* = (x_z^*, y_i)$ for every $\epsilon > 0, k \in N$ satisfying

$$l_\theta(x_z^*, y_i) - \lambda_j \cdot c(z_i^*, (x_i, y_i)) \geq \sup_{z=(x_z, y_z) \in \mathcal{X} \times [L]} l_\theta(x_z, y_z) - \lambda_j \cdot c(z, (x_i, y_i)) - \epsilon$$

$$\text{if} \sup_{z=(x_z, y_z) \in \mathcal{X} \times [L]} l_\theta(x_z, y_z) - \lambda_j \cdot c(z, (x_i, y_i)) < \infty$$

$$l_\theta(x_z^*, y_i) - \lambda_j \cdot c(z_i^*, (x_i, y_i)) \geq k$$

$$\text{if} \sup_{z=(x_z, y_z) \in \mathcal{X} \times [L]} l_\theta(x_z, y_z) - \lambda_j \cdot c(z, (x_i, y_i)) = \infty$$

Let $\mathcal{M}(\xi_1, \xi_2) = \frac{1}{N} \sum_{i \in [N]} \delta(z_i^*, (x_i, y_i))$, and for arbitrariness of $\epsilon$ or $k$, the equality is established.

Finally, we prove the strong duality. Let $\hat{P}_j = P_{N_{LT,j}}(x|y)$, so $W_c(\hat{P}_j, P_{N_{LT,j}}(x|y)) = 0$. Therefore, $\{0 \cdots 0\} \in \mathbb{R}^L$ is the inner point of the set $\{b \in \mathbb{R}^L | b_j = \frac{1}{N_j} \sum_{y_i=j} \int \|\xi - x_i\|_2 Q(d\xi), Q \text{ is a probability measure on } \mathcal{X}\}$, satisfying the slater's condition of a standard conic programming duality result ((Shapiro, 2001), 3.4), thus the strong duality is established. $\qquad \square$

### A.3   Theorem 9 and proof

To analyze the convergence of DRA as in Theorem 9, one needs an assumption on convexity of the cost function $c$, on which the estimation of convergence rate on inner optimization counts. Our choice of the cost function(i.e. $\|x_1 - x_2\|_2^2 + \infty \cdot \mathbf{1}_{\{y_1 \neq y_2\}}$) naturally satisfies the assumption.

**Assumption 8.** The cost function $c(z_1, z_2) := c_x(x_1, x_2) + \infty \cdot \mathbf{1}_{\{y_1 \neq y_2\}}$ satisfies that $c_x(\cdot, x_0)$ is 1-strongly convex for any $x_0$.

**Theorem 9.** *Under similar conditions to WRM (Assumption 8, 10, 12,13). Let $T$ be the number of iterations in outer optimization, $\Delta_F = F(\theta_0) - \inf_{n \in [T]} F(\theta_n)$, $N_{batch}$ is the batchsize, with stepsize of outer optimization $\alpha = \min\{\frac{1}{2L_\Phi}, \sqrt{\frac{N_{batch} \Delta_F}{L_\Phi T \sigma^2}}\}$ and inner stepsize $\alpha_{inner} = \frac{1}{\lambda - L_{zz}}, L_\phi = L_{\theta\theta} + \frac{L_{z\theta} L_{\theta z}}{\lambda - L_{zz}}$ with $L_{\theta z}, L_{\theta\theta}, L_{z\theta}, L_{zz}, \sigma$ depending on $l$. For Algorithm 1, let $s = M - k$ is the number of examples, $\lambda = \min_{j \in [L]} \lambda_j, G = \max\{L_{\theta z}, L_{\theta\theta}, L_{z\theta}, L_{zz}\}$, then for $\{\theta_j\}_{j=0}^T$ in the outer optimization, we have*

$$\frac{1}{T} \sum_{t \in [t]} E[\|\nabla_\theta F(\theta_t)\|_2^2] < \frac{1}{sN_{batch}} \frac{1 - (1 - \frac{\lambda - L_{zz}}{G})^s}{\frac{\lambda - L_{zz}}{G}} \frac{6L_{\theta z}^2}{\lambda - L_{zz}} \epsilon + 4\sigma \sqrt{\frac{1}{N_{batch}} \frac{L_\Phi \Delta_F}{T}}. \tag{16}$$

*Proof.* With similar mild assumptions as WRM (Sinha et al., 2017) below, one can prove that our Distributionally Robust Augmentation Algorithm 1 enjoys a guarantee of convergence. It finds a $\epsilon$-stationary point of model parameters $\theta$ with $O(1/\epsilon^2)$ iterations (as fast as SGD (Amari, 1993)).

**Assumption 10.** The loss $l : \theta \times \mathcal{Z}(\equiv \mathcal{X} \times [L]) \to \mathbb{R}$ satisfies the smoothness conditions:

$$\left\|\nabla_\theta l(\theta, z) - \nabla_\theta l(\theta', z)\right\|_* \leq L_{\theta\theta} \|\theta - \theta^*\|, \quad \left\|\nabla_z l(\theta, z) - \nabla_z l(\theta, z')\right\|_* \leq L_{zz} \|z - z^*\|,$$

$$\left\|\nabla_\theta l(\theta, z) - \nabla_\theta l(\theta, z')\right\|_* \leq L_{\theta z} \|z - z^*\|, \quad \left\|\nabla_z l(\theta, z) - \nabla_z l(\theta', z)\right\|_* \leq L_{z\theta} \|\theta - \theta^*\|.$$

$\|\cdot\|_*$ is the dual norm of the norm $\|\cdot\|$ (as our settings, both are $\|\cdot\|_2$).

We also borrow a lemma from WRM as follows:

**Lemma 11.** *((Sinha et al., 2017), Lemma 1) Let $f : \theta \times \mathcal{Z} \to \mathbb{R}$ be differentiable and $\eta$-strongly concave in $z$, define $\overline{f}(\theta) = \sup_{z \in \mathcal{Z}} f(\theta, z)$. Let $g_\theta(\theta, z) = \nabla_\theta f(\theta, z)$ and $g_z(\theta, z) = \nabla_z f(\theta, z)$ and $f$ satisfies Assumption 10 with replacing $l$ with $f$. Then $\overline{f}$ is differentiable and $\nabla_\theta \overline{f} = \nabla_\theta f(\theta, z^*(\theta))$ where $z^*(\theta) = \sup_{z \in \mathcal{Z}} f(\theta, z)$. And we have*

$$\|z^*(\theta_1) - z^*(\theta_2)\| \leq \frac{L_{z\theta}}{\eta} \|\theta_1 - \theta_2\|,$$

*and*

$$\left\| \nabla \overline{f}(\theta) - \nabla \overline{f}(\theta^*) \right\|_* \leq (L_{\theta\theta} + \frac{L_{z\theta}L_{\theta z}}{\eta}) \left\| \theta_1 - \theta_2 \right\|.$$

Under Assumption 10, the primal of inner optimization $l(\theta, z) - \lambda_j c(z, z_0)$ is $(\lambda_j - L_{zz})$-strongly concave, so that $z^*_{j,0} = \arg\sup_z l(\theta, z) - \lambda_j c(z, z_0)$ is well-defined and satisfies the condition of Lemma 11. Thus, we have $\nabla_\theta \sup_{z\in\mathcal{Z}} l(\theta, z) - \lambda_j c(z, z_0) = \nabla_\theta l_\theta(z^*_{j,0})$ and $\nabla_\theta F(\theta)$ is $\frac{L_{z\theta}L_{\theta z}}{\lambda_j - L_{zz}}$-Lipschitz. Then we make an assumption on the variance of the gradient in the outer-optimization, which is a common condition when analyzing the convergence of SGD.

**Assumption 12.** For any sampled training set $\{(x_i, y_i)\}_{i\in[N]}$, it holds

$$E \left\| \nabla_\theta l_\theta(z^*_{y_i,i}) - \nabla_\theta F(\theta) \right\|^2 \leq \sigma^2,$$

where $z^*_{y_i,i} = \arg\sup_z l_\theta(z) - \lambda_j c(z, (x_i, y_i))$.

With the preparation above, we begin to prove Theorem 9. Let $h_j(\theta, z; (x_i, j)) := l(\theta, z) - \lambda_j c(z, (x_i, j))$. In Algorithm 1, the gradient we use to update is actually

$$g_t = \frac{1}{N_{batch}} \sum_{i\in[N_{batch}]} \frac{1}{s} \sum_{l\in[s]} \nabla_\theta h_{y_i}(\theta_t, (x_i^{l+k}, y_i); (x_i, y_i)).$$

Since the gradient of $F(\theta)$ is $L_\phi$-smooth,

$$F(\theta_{t+1}) < F(\theta_t) + \langle \nabla F(\theta_t), \theta_{t+1} - \theta_t \rangle + \frac{L_\phi}{2} \left\| \theta_{t+1} - \theta_t \right\|_2^2$$

$$= F(\theta_t) - \alpha(1 - \frac{L_\phi\alpha}{2}) \left\| \nabla F(\theta_t) \right\|_2^2 + \alpha(1 + \frac{L_\phi\alpha}{2}) \langle \nabla F(\theta_t), F(\theta_t) - g_t \rangle$$

$$+ \frac{L_\phi\alpha^2}{2} \left\| F(\theta_t) - g_t \right\|_2^2.$$

We need to estimate $F(\theta_t) - g_t$ in the equation above. Let

$$g_t^* = \frac{1}{N_{batch}} \nabla_\theta \sum_{i\in[N_{batch}]} l(\theta_t, z^*_{y_i,i}(\theta_t)).$$

With the help of $g_t^*$, we can estimate $F(\theta_t) - g_t$ more easily. The difference between $g_t$ and $g_t^*$ comes from we use a series of points to compute the gradient to $\theta$ and the inner optimization cannot obtain an optimal solution precisely. The difference between $\nabla F(\theta_t)$ and $g_t^*$ is from just sampling a batch from all training instances.

We compute the difference between $g_t$ and $g_t^*$ as $\delta_t = g_t - g_t^*$, then we have

$$\|\delta_t\|_2^2 = \left\| \frac{1}{N_{batch}} \sum_{i\in N_{batch}} \{ \frac{1}{s} \sum_{l\in[s]} \nabla_\theta l(\theta_t, (x_i^{l+k}, y_i)) - \nabla_\theta l(\theta_t, z^*_{y_i,i}) \} \right\|_2^2$$

$$\leq (\frac{1}{N_{batch}})^2 \sum_{i\in N_{batch}} (\frac{1}{s})^2 \sum_{l\in[s]} l_{\theta z}^2 \left\| (x_i^{l+k}, y_i) - z^*_{y_i,i} \right\|_2^2.$$

**Assumption 13.** The inner-optimization just reaches the linear convergence rate (Nesterov, 1998) of gradient descent in strongly-convex optimization i.e. $\|(x_i^{k+s}, y_i) - z^*_{y_i,i}\|_2^2 \leq (1 - \frac{\lambda - L_{zz}}{G})^{k+s} \|(x_i^0, y_i) - z^*_{y_i,i}\|_2^2 \leq \epsilon.$

We make this assumption under the motivation that the last example is enough when the convergence is much faster than the convergence bound that the strongly-convex condition gives. However when the convergence of inner-optimization is slow i.e. just reaches the convergence bound, using a sequence of examples can gain benefits.

We have $\left\|(x_i^{k+s}, y_i) - z_{y_i,i}^*\right\|_2^2 \leq \frac{1}{\lambda_{y_i} - L_{zz}} \epsilon$. With another assumption that inner-optimization reaches the linear convergence rate of gradient descent on strongly concave optimization (Nesterov, 1998), as the inner optimization is at least $\lambda - L_{zz}$-strongly concave, we have

$$\left\|(x_i^{k+l}, y_i) - z_{y_i,i}^*\right\|_2^2 \leq (1 - \frac{\lambda - L_{zz}}{G})^{s-l} \frac{1}{\lambda_{y_i} - L_{zz}} \epsilon.$$

As a result, $\|\delta_t\|_2^2 \leq \frac{1}{sN_{batch}} \frac{1 - (1 - \frac{\lambda - L_{zz}}{G})^s}{\frac{\lambda - L_{zz}}{G}} \frac{4L_{\theta z}^2}{\lambda - L_{zz}} \epsilon$.

Substituting $g_t$ with $\delta_t$, we have

$$F(\theta_{t+1}) < F(\theta_t) - \frac{\alpha}{2} \|\nabla F(\theta_t)\|_2^2 + \frac{\alpha}{2}(1 - \frac{1}{2}L_\phi \alpha) \|\delta_t\|_2^2$$
$$+ \alpha(1 - L_\phi \alpha)\langle \nabla F(\theta_t), \nabla F(\theta_t) - g_t^* \rangle + L_\phi \alpha^2 (\|\delta_t\|_2^2 + \|\nabla F(\theta_t) - g_t^*\|_2^2).$$

For Lemma 11, $g_t^*$ is unbiased estimation to $\nabla F(\theta_t)$ so $E[g_t^* - \nabla F(\theta_t)|\theta_t] = 0$. Using Assumption 12 to control the invariance of the estimation and taking expectations of the above formula, we have

$$E[F(\theta_{t+1}) - F(\theta_t)] \geq -\frac{\alpha}{2} E[\|\nabla F(\theta_t)\|_2^2]$$
$$+ \frac{1}{N_{batch}} L_\phi \alpha^2 \sigma^2 + \frac{3}{4}\alpha \frac{1}{sN_{batch}} \frac{1 - (1 - \frac{\lambda - L_{zz}}{G})^s}{\frac{\lambda - L_{zz}}{G}} \frac{4L_{\theta z}^2}{\lambda - L_{zz}} \epsilon,$$

where we use $\alpha < \frac{1}{2L_\phi}$ to get $\frac{3}{2}L_\phi \alpha^2 \leq \frac{3}{4}\alpha$.

Summing by $t$, we get

$$\frac{1}{T} \sum_{t=1}^{T} E[\|\nabla F(\theta_t)\|_2^2] \leq 2\frac{\Delta F}{\alpha T} + \frac{2}{N_{batch}} L_\phi \alpha \sigma^2 + \frac{6}{sN_{batch}} \frac{1 - (1 - \frac{\lambda - L_{zz}}{G})^s}{\frac{\lambda - L_{zz}}{G}} \frac{4L_{\theta z}^2}{\lambda - L_{zz}} \epsilon.$$

Substituting with the stepsize $\alpha$, the conclusion is obtained. $\qquad\square$

*Remark* 1. Actually, in the inequality (16), considering that $\lambda_j$ is small but concavity is established ($\lambda_j \geq L_{zz}$), we assume $\frac{\lambda_j - L_{zz}}{G} < 1$, then the first term of the bound reduces with the increase of the number of examples $s$ thus the overall optimization becomes more stable.

*Remark* 2. Previous WRM can be regarded as using the last point to compute gradient for outer-optimization, since if inner-optimization is concave with small radius (corresponding to big $\lambda_j$), it is the optimal solution, which leads to unbiased estimation for the gradient of outer optimization(Sinha et al., 2017). Considering the situation that with small $\lambda_j$, the inner optimization is relatively slow and concavity may even not be established. Therefore we propose a more moderate way by using a sequence of examples from the iterations of inner-optimization instead of making a locally optimal point the only example. Theorem 9 gives evidence to our above insight.

## A.4   Generalization bound and optimality of augmentation examples

In addition, we give a bound on the estimated balanced risk on real distribution and the optimality of examples generated by DRA under mild conditions.

**Theorem 14** (Generalization bound). *Assuming $l : \Theta \times (\mathbb{R}^m \times [L]) \to \mathbb{R}$ is continuous and for every class $j$, let the real class-conditional distribution $P(x|y_j)$ satisfies $\exists \alpha > 0$ and $x_0 \in \mathbb{R}^m$, $A = E_{P(x|y_j)}\{c_x(x, x_0)^\alpha\} < \infty$ with $c_x$ a valid cost function, then $\exists c_1, c_2$ only depending on $A, \alpha$, for any multiplier $\{\lambda_j\}_{j \in [L]}$, with probability $1 - \eta$ it holds*

$$R_{bal,l} \leq \sum_{j \in [L]} \frac{1}{L}\{\lambda_j \max\{(\frac{log(c_1/\eta)}{c_2 N_j})^{1/m}, (\frac{log(c_1/\eta)}{c_2 N_j})^{1/\alpha}\}$$
$$+ E_{P_{N,LT}(x|y_j)}[\sup_{z=(x_z, y_z) \in \mathcal{X} \times [L]} l(x_z, y_z) - \lambda_j \cdot c(z, (x, y))]\},$$

*in which $R_{bal,l} := \sum_{j \in [L]} \frac{1}{L} E_{P(x|y_j)}\{l(x,y)\}$ is the balanced risk on test distribution $P_{test} = \frac{1}{L} \sum_{j \in [L]} P(x|y_j)$ estimated by loss function $l(x,y)$.*

*Proof.* Using Theorem 7, for any multiplier $\{\lambda_j\}_{j \in [L]}$, we have

$$\frac{1}{L} \sum_{j \in [L]} \sup_{\hat{P}_j \in \mathcal{P}_j} E_{\hat{P}_j}\{l(x,y)\} < \frac{1}{L} \sum_{j \in [L]} \lambda_j r_j$$
$$+ E_{P_{N,LT}(x|y_j)}[\sup_{z=(x_z,y_z) \in \mathcal{X} \times [L]} l(x_z,y_z) - \lambda_j \cdot c(z,(x,y))],$$

in which $r_j$ is the radius of $\mathcal{P}_j$.

And with our assumption, $P(x|y_j)$ satisfies conditions of Lemma 6. So let $r_j = \max\{(\frac{log(c_1/\eta)}{c_2 N_j})^{1/m}, (\frac{log(c_1/\eta)}{c_2 N_j})^{1/\alpha}\}$. By applying Lemma 6, with the probability of $1 - \eta$, the distance $W_c(P(x|y_j), P_{N,LT}(x|y_j)) < r_j$ and we have $P(x|y_j) \in \mathcal{P}_j = \{\hat{P}_j | W_c(\hat{P}_j, P_{N,LT}(x|y_j)) < r_j\}$. Thus, it establishes

$$\sum_{j \in [L]} \frac{1}{L} E_{P(x|y)}\{l(x,y)\} < \frac{1}{L} \sum_{j \in [L]} \sup_{\hat{P}_j \in \mathcal{P}_j} E_{\hat{P}_j}\{l(x,y)\}$$

and we get the conclusion. $\square$

*Remark* 3. This bound shows with a high probability the balanced risk on real distribution could be bounded by our objective (the last term in the right) plus a constant depending on the multiplier $\lambda_j$ and the number of instances from each class, which serves as a generalization gap, decreasing with the number of instances $N_j$. It indicates $\lambda_j$ controls a trade-off between robustness and performance. If we do not enforce any robustness i.e. the multipliers $\lambda_j$ are extremely large, DRA actually equals to ERM. In this situation, large $\lambda_j$ leads to considerable generalization gap, especially for the tail classes. Hence, **we conclude that models robust to CCD shift benefit long-tailed recognition.** Conversely, making $\lambda_j$ extremely small near zero, it admits a trivial bound meaning the model refuses to make predictions, which is well-known as "over-pessimism" in DRO (Frogner et al., 2021).

The above bound also shows a new viewpoint beyond existing theoretical results of long-tailed recognition in the community. We make some remarks for the bound as follows:

1. It is more flexible and tight than the results of prior DRO works (Sinha et al., 2017; Kuhn et al., 2019). It adjusts class-wise robustness by $\lambda_j$ for each class $j$, while prior work only admits a special case of $\lambda_j = C, for\,all\,j \in [L]$.

2. It is closer to practical training compared to the Fisher-consistency result by logit adjustment (Menon et al., 2020). Fisher-consistency theory from Theorem 1 in (Menon et al., 2020) only states that a Bayes-optimal classifier would be obtained by the minimization of logit adjustment loss under a balanced label distribution with real CCDs.

3. It is no longer dependent on the capacity of the hypothesis class, which is extremely large for modern neural networks, while the bound in Theorem 1 of (Cao et al., 2019) is.

**Proposition 15** (Optimality of augmentation examples). *Assuming $l : \Theta \times (\mathbb{R}^m \times [L]) \to \mathbb{R}$ is continuous and Assumption 8 is satisfied. $T_j((x,y)) := arg\max_{z=(x_z,y_z) \in \mathcal{X} \times [L]} l(x_z,y_z) - \lambda_j \cdot c(z,(x,y))$ is the optimal solution of inner optimization, $\mathcal{M}$ is the set of probability measure on $\mathcal{X} \times [L]$ and $P^*_{N,LT}(x|y_j) = \frac{1}{N_{LT,j}} \sum_{\{i:y_i=y_j\}} \delta(T_j((x_i,y_i)))$ is the empirical distribution consisting of them. Then for any multiplier $\{\lambda_j\}_{j \in [L]}$, we have*

$$P^*_{N,LT}(x|y_j) = \arg\max_{\hat{P}_j \in \mathcal{M}} E_{\hat{P}_j}[l(x,y)] - \lambda_j W_c(\hat{P}_j, P_{N,LT}(x|y_j)),$$

*which indicates the empirical distribution consisting of ideal optimal solutions of inner optimization of a class is the optimal perturbed distribution of the Lagrange penalty problem.*

*Proof.*

$$\sup_{\hat{P}_j \in \mathcal{M}} E_{\hat{P}_j}[l(x,y)] - \lambda_j W_c(\hat{P}_j, P_{N,LT}(x|y_j))$$

$$= E_{P_{N_{LT},j}(x|y)}[\sup_{z=(x_z,y_z)\in\mathcal{X}\times[L]} l(x_z,y_z) - \lambda_j \cdot c(z,(x,y))]$$

$$= E_{P^*_{N,LT}(x|y_j)}[l(x,y)] - \lambda_j \sum_{\{i:y_i=y_j\}} c(T_j((x_i,y_i)),(x_i,y_i))$$

$$\leq E_{P^*_{N,LT}(x|y_j)}[l(x,y)] - \lambda_j W_c(P^*_{N,LT}(x|y_j), P_{N,LT}(x|y_j))$$

The first equality is from Theorem 7 and the second is from the definition of $P^*_{N,LT}(x|y_j)$. The inequality is from the definition of Wasserstein distance. For the inequality in the opposite direction is trivial, we establish the conclusion. □

### A.5 Proof of 17 in the toy example

*Proof.*

$$E_{P_N(x|c)}[l(x,y)] = -\frac{1}{N} \sum_{i\in[N_c]} log(\frac{e^{\kappa\mu_c^T x}}{Z})$$

$$= log(Z) - \kappa\mu^T \hat{x}^c$$

while $E_{P(x|c)}[l(x,y)]$ is the entropy of vmF distribution $-log(Z) - \kappa A(\kappa)$(Sra, 2012). And the conclusion is established. □

## B Detailed Empirical Study on CCD Shift

### B.1 A toy example to validate CCD shift

In Sec 3, we argue that even with Fisher-consistent loss function, loss computed on empirical distribution with scarce instances is unreliable to estimate the risk on real test distribution, as Proposition 1 indicates, exhibiting the shift between empirical CCD and the real CCD. To further clarify the concept of CCD shift, we visualize and analyze a toy example.

We assume that there exists data from three classes with imbalanced ratio of $10:5:1$, sampled from an ideal feature space, i.e. the feature $x$ is located on 3-sphere: $S^2 = \{x \in \mathbb{R}^3 | \|x\| = 1\}$. $\mu_c$ is the feature prototype of class $c$, located on the vertices of a regular simplex i.e. uniformly distributed on the hypersphere (Li et al., 2022c; Graf et al., 2021). For each class $c \in \{0,1,2\}$, the instances are sampled from the distribution $P(x|c)$, a vMF distribution(Banerjee et al., 2005) on the $S^2$ with the density $p(x|c) = \frac{1}{Z}e^{\kappa\mu_c^T x}$. $Z$ is the normalization constant $\frac{2\pi(e^\kappa - e^{-\kappa})}{\kappa}$. Specifically, we set $\kappa = 1$ and sample $N_2 = 10$ instances for class 2.

Figure 8 visualizes the sampled instances $\{x_i^c\}_{i=1}^{N_c}$, their normalized mean projected to $S^2$ $\frac{\hat{x}^c}{\|\hat{x}^c\|}$ with $\hat{x}^c = \frac{1}{N_c}\sum_{i=1}^{N_c} x_i^c$ and the prototype $\mu_c$ For class 2 that has fewest instances, the normalized mean of the sampled instances $\frac{\hat{x}^2}{\|\hat{x}^2\|}$ deviates from the prototype $\mu_2$,with cosine similarity of 0.7424, significantly lower than similarities of other two classes. These indicates the empirical distribution of the class with scarce instances is not a reliable estimation to the real distribution of the class, i.e. CCD shift.

Furthermore, one can check the shift more formally, by computing the value of a specific loss function on the empirical CCD or real CCD. Taking $l(x,y) = -\sum_{c\in[3]} \mathbf{1}_{\{y=c\}} log(p(x|c))$ as the loss function, the difference between empirical CCD $P_N(x|c) = \frac{1}{N_c}\sum_{i\in[N_c]} \delta(x_i^c)$ and real CCD $P(x|c)$ is

$$\left|E_{P(x|c)}[l(x,y)] - E_{P_N(x|c)}[l(x,y)]\right| = \kappa\left|\mu_c^T \hat{x}^c - A(\kappa)\right| \tag{17}$$

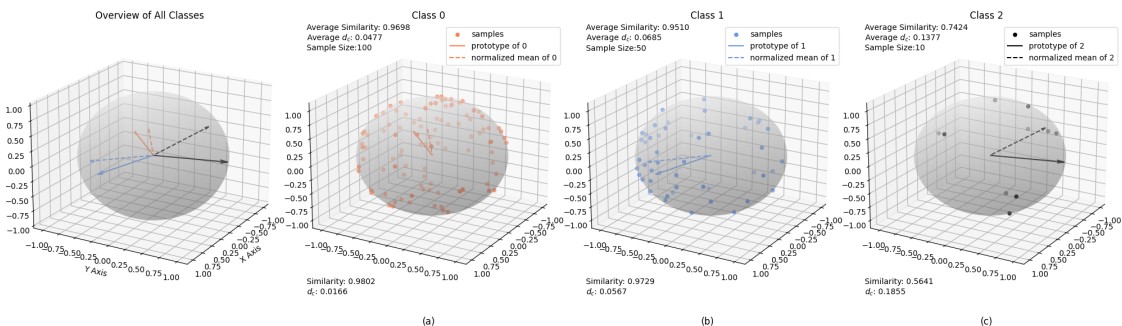

Figure 8: Illustration of CCD shift by a toy example. Subfigures (a), (b) and (c) correspond to class 0, 1, and 2. We report average similarity and loss difference between empirical CCD and real CCD $d_c$ on 50 runs. The visualization is from one run and its similarity and loss difference are also reported. The estimated average mean of tail class 2 deviates from the prototype significantly, along with less similarity and larger $d_c$, indicating the unreliability of empirical CCD from scarce instances.

noted as $d(c; \kappa)$, where $A(\kappa) = \frac{J_{\frac{3}{2}}(\kappa)}{J_{\frac{1}{2}}(\kappa)}$ is a constant and $J_n(x)$ is Bessel functions of the first kind (Sra, 2012). See the derivation of (17) in Appendix A.5. As in Figure 8, the tail class 2 exhibits much more difference $d_c$ than class 0, 1 with more instances, acting as en evidence to Proposition 1. Empirical CCD cannot estimate real CCD reliably for classes with few instances, therefore CCD shift cannot be disregarded in long-tailed recognition.

## B.2 Removing shift sampling

**Detailed description of removing shift sampling.** Many benchmarks of long-tail recognition, such as CIFAR10/100-LT (Cao et al., 2019) and ImageNet-LT (Liu et al., 2019), are down-sampled from balanced datasets. Formally, instances from balanced datasets and long-tailed dataset are sampled respectively from

$$P_{N,bal}(x,y) = \sum_{j \in [L]} P_{bal}(y_j) P_{N_j,bal}(x|y_j), \tag{18}$$

$$P_{N,LT}(x,y) = \sum_{j \in [L]} P_{LT}(y_j) P_{N_j,LT}(x|y_j). \tag{19}$$

Here $P_{N_j,bal}(x|y_j) = \frac{1}{N_{bal,j}} \sum_{\{i:y_i=y_j\}} \delta(x_i, y_i)$ and $P_{N_j,LT}(x|y_j) = \frac{1}{N_{LT,j}} \sum_{\{i:y_i=y_j\}} \delta(x_i, y_i)$. Besides, $P_{bal}(y)$ is uniform and $P_{LT}(y)$ is imbalanced. To conduct an ablation comparison with respect to CCD, we replace the CCD in (19) with $P_{N_j,bal}(x|y_j)$, which is more reliable by Proposition 1, as an oracle for real CCD. While $P_{N_j,bal}(x|y_j)$ may not approximate real CCD well, as long as the approximation brings significant improvement on performance (as in our experiments and see more in next section Appendix B.2), we can identify CCD shift according to Proposition 1. That is, sampling from the following distribution

$$P_{remove\ shift}(x,y) = \sum_{j \in [L]} P_{LT}(y_j) P_{N_j,bal}(x|y_j),$$

which is our proposed *removing shift sampling* in Section 3.2.

**Assumption on removing shift.** To conduct experimental comparisons on CCD, we propose removing shift sampling in Section 3.2. An ideal situation for comparison on CCD is to obtain real CCD $P(x|y)$ and keep imbalanced label distribution $P(y)$. In fact, due to no access to the real $P(x|y)$, we make **the empirical CCD $P_{N,bal}(x|y)$ serves as an oracle to substitute real CCD $P(x|y)$**. At the same time, we keep imbalance by using the same label distribution $P_{LT}(y)$ as long-tailed datasets so that the label distribution is imbalanced in every batch and so it is in the whole training. In this way, more unique instances are seen by the model but the number of samples from each class, i.e. label distribution, keeps unchanged in all batches

during training. So under our assumption, we make a fair ablation study on CCD with proper control of other factors. It is possible that $P_{N,bal}(x|y)$ cannot estimate real CCD well, but as long as the oracle brings improvement in performance, we can blame CCD shift correctly. The counterexample appears in extreme cases when the oracle does not make improvement e.g. using re-weighting with removing shift sampling in the training process, where the removing shift sample does not bring improvement. In that case, we cannot tell whether the CCD shift does not appear thus should not be blamed or our oracle is too weak to identify the CCD shift.

### B.3 Experimental setting

We perform ablation experiments on CIFAR-LT without or with removing shift sampling to investigate the effect of CCD shift on vanilla ERM (Vapnik, 1991) and some representative re-balancing methods: DRW Cao et al. (2019), CRT (Kang et al., 2019) and PC softmax (Menon et al., 2020; Hong et al., 2021).

**Implementation of removing shift sampling.** We do removing shift sampling on original balanced dataset CIFAR10/100 (Krizhevsky, 2009). We use a dataloader with a class-imbalanced sampler which samples with the same class probability $P_{LT}(y)$ as the long-tailed dataset and samples uniformly within each class. The accumulated probabilities of instances from each class equals the predetermined class frequency and every instance within a class has the same sampling probability. To this end, we implement the sampler as a multinomial distribution sampler in the same way as that used to generate CIFAR-LT in Cao et al. (2019). Different from the generation of long-tailed dataset, we sample from the whole balanced dataset and get class-imbalanced instances in every batch. In addition, We keep the same amount of data as the long-tailed dataset while only change the class-conditional distribution via removing shift in the ablation experiment.

**More implementation details.** We train ResNet-32 (He et al., 2016) with batch size 256, optimized by SGD with momentum 0.9 and weight decay $5 \times 10^{-4}$ with warm-up scheduler, consistent with the setting of our experiments on DRA. The learning rate decays by a factor of 0.01 at epochs 160 and 180 with an initial rate 0.2. We evaluate Top-1 accuracy on the original validation set of the datasets, following the common protocol in long-tailed recognition (Cao et al., 2019).

**Methods in experiments.** We select ERM as baseline, and DRW, PC softmax (equals to post-hoc logit adjustment) and CRT as representative re-balancing methods.

1. **ERM.** ERM means empirical risk minimization (Vapnik, 1991). We train a model on long-tailed datasets with cross-entropy loss without any strategy.

2. **Balance.** For easy reference, we also train a model under our setting on the original CIFAR10/100.

3. **DRW.** We implement DRW (Cao et al., 2019) following the original paper. We train in a regular way in the early phase and re-weight the loss by the inverse of the predetermined class frequencies only in the last phase during training. We typically re-weight the loss function starting at 160 epochs.

4. **RW.** RW means directly re-weighting the loss by the inverse of the predetermined class frequencies. We re-weight the loss in the whole training phase. In ablation experiments on decoupling methods, we utilize DRW and RW to obtain re-balanced features and ERM to obtain imbalanced features in the first stage of training.

5. **PC softmax.** We implement PC softmax following (Hong et al., 2021) by training the model by ERM for 200 epochs and applying adjustment at testing.

6. **CRT.** We implement CRT following (Kang et al., 2019). We train the whole model for 200 epochs and re-train the classifier for 10 epochs with re-weighting loss. We restart the learning rate when re-training the classifier.

7. **LWS.** We implement LWS following (Kang et al., 2019). We assign weights to classifiers with learnable scale factors i.e. $\hat{W}_i = f_i \cdot W_i, i \in [L]$. We train the whole model for 200 epochs and

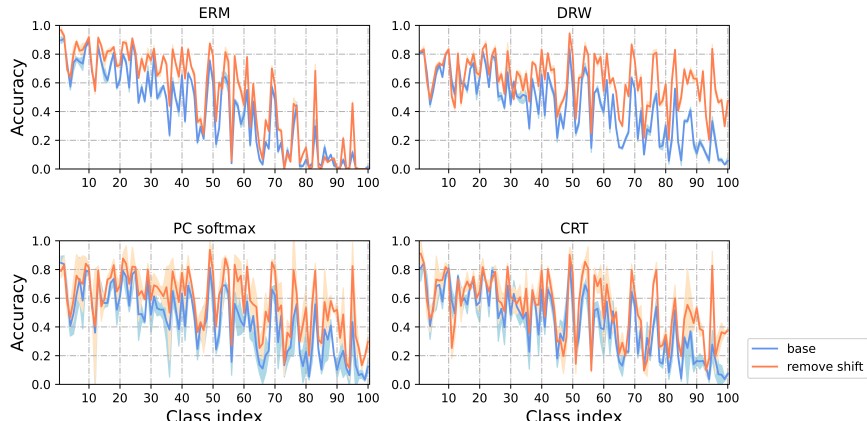

Figure 9: Accuracy on CIFAR100-LT without or with removing CCD shift. It shows similar results that models trained with removing shift sampling get significant performance improvement.

Table 8: Accuracy of different methods on CIFAR-LT without or with removing CCD shift

|  | CIFAR100-LT | | CIFAR10-LT | |
|  | removing shift | base | removing shift | base |
|---|---|---|---|---|
| ERM | 50.7 | 40.28 | 80.37 | 73.24 |
| DRW | 63.97 | 44.41 | 89.01 | 78.04 |
| PC softmax | 58.61 | 44.34 | 87.90 | 79.58 |
| CRT | 52.82 | 43.48 | 87.07 | 77.72 |
| Balance | 66.73 | | 91.53 | |

then train factors $f_i$ for 10 epochs by re-weighting the loss. We tune the learning rate on different datasets when learning scale factors as we found it is somewhat sensitive.

## B.4 More results and analysis on CIFAR100-LT

On a whole, results on CIFAR100 agree with those on CIFAR10-LT. In Figure 9 and Table 8, removing shift sampling improves the performance of re-balancing methods and ERM on CIFAR100-LT, which indicates again that shift of CCDs is the key to limit long-tailed recognition.

As for decoupling methods, Table 9 shows re-balanced features outperforms those uniformly trained significantly on two decoupling methods CRT and LWS. It agrees with our explanation of why decoupling methods work: first-stage learning without re-balancing avoids more severe CCD shift while re-balancing with removing shift could benefit feature learning.

Results of logit adjustment on CIFAR100-LT in shown Figure 10. With removing shift sampling $\tau = 1$ is optimal, while without it the best $\tau$ is much bigger than 1. This result again agrees with our supposition that logit adjustment gets sub-optimal in experiments and leaves space to be improved due to CCD shift.

## B.5 Confidence calibration and feature deviation under CCD shift

As two recently studied issues in long-tailed recognition, confidence calibration (Guo et al., 2017) and feature deviation (Ye et al., 2020) have drawn an amount of attention. We conduct experiments to investigate how the shift of CCDs affects these two issues.

**Confidence calibration.** Confidence calibration is to make model prediction estimate true correctness likelihood well, which is important in many applications. It has been proved that the calibration of neural

Table 9: Accuracy of decoupling methods, CRT and LWS, on CIFAR-LT with different features, without or with removing shift. LWS shows similar results to CRT: re-balance harms feature learning while benefits feature learning with removing shift.

| feature re-balance | classifier adjust | CIFAR100-LT | | CIFAR10-LT | |
|---|---|---|---|---|---|
| | | removing shift | base | removing shift | base |
| - | CRT | 52.82 | 43.48 | 87.07 | 77.72 |
| DRW | CRT | 58.39+5.57 | 42.36-1.12 | 88.42+1.35 | 75.97-1.75 |
| - | LWS | 57.67 | 44.05 | 87.55 | 76.21 |
| DRW | LWS | 61.95+4.28 | 43.63-0.42 | 88.95+1.40 | 75.36-0.84 |
| DRW | | 63.97 | 44.41 | 89.01 | 78.04 |
| RW | CRT | 48.11-4.71 | 28.48-15.0 | 79.51-7.56 | 71.43-6.29 |
| RW | LWS | 49.03-8.64 | 29.76-14.29 | 78.75-8.80 | 71.78-4.43 |
| RW | - | 49.92 | 30.48 | 80.34 | 72.78 |
| Balance | - | 66.73 | | 91.53 | |

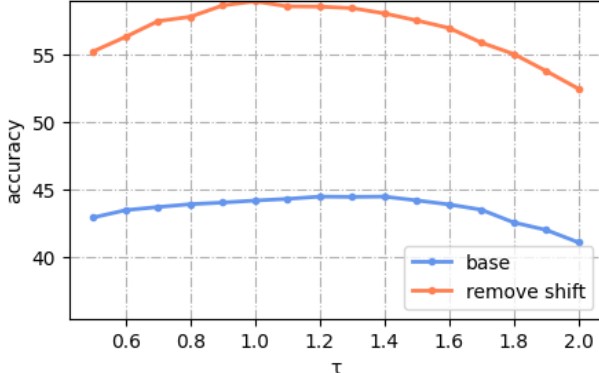

Figure 10: Accuracy on CIFAR100-LT with varying $\tau$ in post-hoc adjustment. After removing shift $\tau = 1$ gets optimal performance similar to that on CIFAR10-LT.

Table 10: ECE on CIFAR-LT of different methods without or with removing shift sampling.

| | CIFAR10-LT | | CIFAR100-LT | |
|---|---|---|---|---|
| | removing shift | base | removing shift | base |
| ERM | 6.4 | 15.5 | 15.91 | 29.75 |
| DRW | 2.00 | 10.77 | 2.19 | 22.16 |
| PC softmax | 2.55 | 9.25 | 2.73 | 19.43 |
| CRT | 4.40 | 14.54 | 19.5 | 26.63 |
| Balance | 1.77 | | 3.89 | |

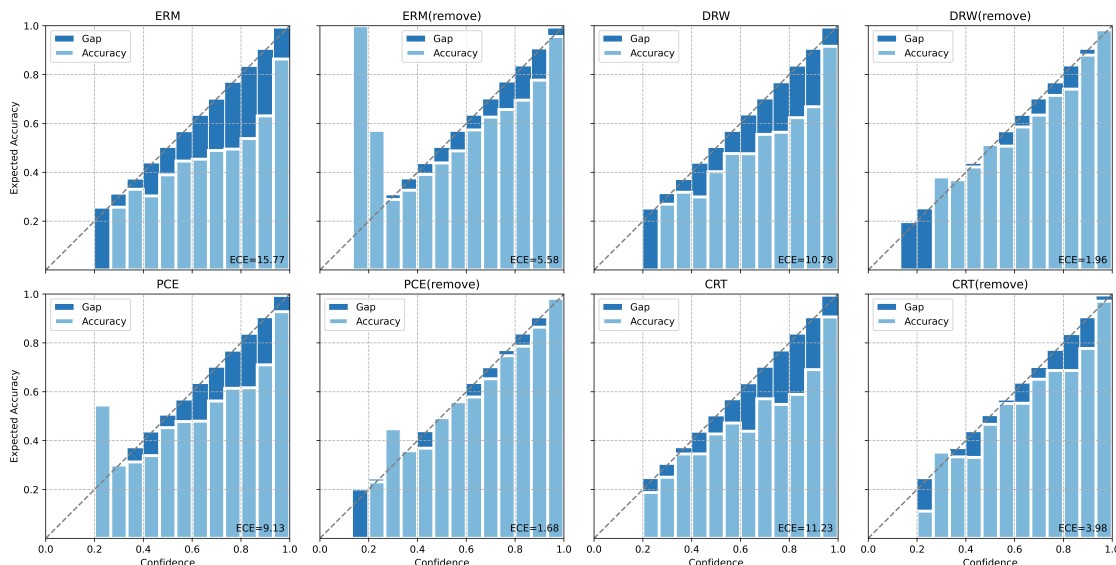

Figure 11: Reliability diagrams of different models trained on CIFAR10-LT. "(remove)" means using removing shift sampling and PCE means PC softmax. The gap between model confidence and error probability is reduced by removing shift sampling significantly.

Table 11: Average feature deviation distance on CIFAR-LT of different methods without or with removing shift sample

|  | CIFAR10-LT | | CIFAR100-LT | |
|---|---|---|---|---|
|  | removing shift | base | removing shift | base |
| ERM | 0.83 | 1.27 | 2.11 | 3.18 |
| DRW | 0.693 | 1.20 | 1.96 | 2.90 |
| Balance | 0.73 | | 2.10 | |

networks is generally bad and long-tailed distributions make neural networks even more miscalibrated and over-confident (Guo et al., 2017; Zhong et al., 2021). *Expected Calibration Error* (ECE) is widely used to measure the calibration of a model. With all $N$ instances divided into $B$ interval bins of equal size by their predictions, ECE is calculated as:

$$ECE = \sum_{i=1}^{B} \frac{|S_i|}{N} |acc(S_i) - conf(S_i)|,$$

in which $S_i$ is the set of instances whose predictions fall into the $i$-th bin. $acc(\cdot)$ and $conf(\cdot)$ compute the the accuracy and estimated confidence on $S_i$ respectively. As shown in Table 10, re-balancing methods improve ECEs except CRT on CIFAR-100 and removing shift sampling improves the calibration of different models significantly. However, using both re-balancing methods and removing shift sampling is still worse (in ECE) than that on balanced dataset. We suppose the reason lies in that we still cannot perfectly model a real $P(x|y)$ even with our sampling method as the whole sample size is the same as down-sampled long-tailed dataset, fewer than original balanced dataset. Figure 11 shows the reliability diagrams of different models trained on CIFAR10-LT. With removing shift sampling, the uncertainty estimation of models is significantly improved.

**Feature deviation.** Feature deviation is a phenomenon found in long-tailed recognition by recent works (Ye et al., 2020; 2021). That is the average distance of features learned from long-tailed training dataset usually exhibits imbalance between training and test and the distances of tail classes are larger than those of

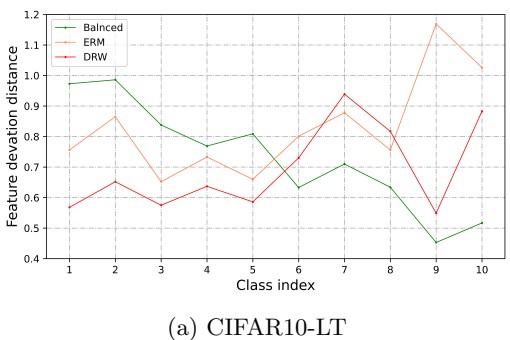

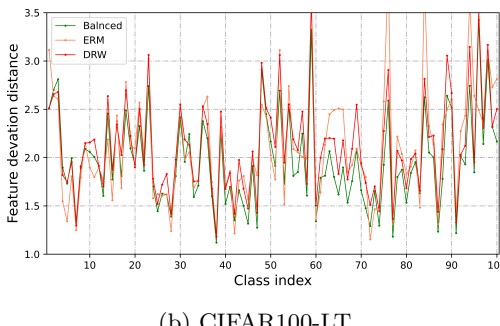

(a) CIFAR10-LT

(b) CIFAR100-LT

Figure 12: Class-wise feature deviation on CIFAR10-LT/CIFAR100-LT with removing shift sampling. Even with removing shift sampling and re-balancing it still exhibits imbalanced feature deviation on both CIFAR10-LT and CIFAR100-LT.

head classes evidently. (Ye et al., 2020) proposes a *feature deviation distance $dis(j)$* to measure the deviation for class $j$:

$$dis(j; g_\theta) = \frac{1}{R} \sum_{i=1}^{R} \left\| mean(S_K(\{g_\theta(x_{train}^{(j)})\})) - mean(\{g_\theta(x_{test}^{(j)})\}) \right\|_2,$$

in which $\|\cdot\|_2$ is L2-norm, $S_K$ means sub-sampling $K$ instances from class $j$, $R$ is the number of sampling rounds, and $g_\theta$ is the feature extractor. And for convenient comparison, we also use a class-wise mean of feature deviation distance, called *average feature deviation distance*:

$$\overline{dis} = \frac{1}{L} \sum_{j \in [L]} dis(j; g_\theta). \tag{20}$$

Table 11 shows feature deviation is affected by shift of CCDs as removing shift sampling improves this metric on the whole while the re-balancing method (i.e. DRW) also generally alleviates the feature deviation no matter with removing shift or not. Besides, training equipped with removing shift seemingly (regardless of re-balance or not) approachs or outperforms the balanced result, e.g. 0.69/0.83 vs 0.829 on CIFAR10-LT and 1.96/2.11 vs 2.10 on CIFAR100-LT. One may conclude that imbalance does not make an obvious difference on feature deviation. In fact, this is because features of head classes benefit from training on imbalance distribution, thus being more compact. As a result, using average feature deviation distance as the measure, re-balance seems not critical for alleviating feature deviation.

Figure 12 shows class-wise feature deviation is affected by imbalance obviously as well. The feature deviation distances of different classes still exhibit imbalance with removing shift and there exists significant gap of feature deviation distance of tail classes between balanced training and re-balanced training with removing shift. We suppose that is because re-balancing methods used in the experiments still leave space to improve feature deviation, e.g. it has been discovered that the imbalanced logit distribution harms feature learning in the beginning stage of training (Ye et al., 2021).

## C   Complete Experiments on DRA

### C.1   Experimental details

**CIFAR100-LT and CIFAR10-LT.** Unless specifically stated, we use the setting from (Hong et al., 2021) on CIFAR10/100-LT by default. Under this setting, we apply SGD with batch size 256 and the base learning rate 0.2 to train a ResNet-32 (He et al., 2016) model for 200 epochs, following (Hong et al., 2021). We employ the linear warm-up learning rate schedule for the first five epochs and reduce it at epochs 160 and 180 by a factor of 0.01 and use the same weight decay 0.0005 as previous works (Cao et al., 2019; Zhou et al., 2022). For the setting of (Zhou et al., 2022) in the Table 14, the batch size and learning rate are changed into 128 and 0.1 respectively, and weight decay is set as 0.0002, keeping the same with (Zhou et al., 2022) to make

fair comparisons with previous works(Kini et al., 2021; Kim et al., 2020; Zhou et al., 2022) as we find their hyperparameters are sensitive with the batchsize and weight decay.

**Tiny-ImageNet-LT.** Following (Park et al., 2021), we apply SGD with batch size 128 and weight decay 0.0002 to train a ResNet-18 model for 100 epochs. We set the base learning rate to 0.1 and reduce it at epochs 50 and 90 by a factor of 0.1. Unless specifically stated, we perform training for 180 epochs by default.

**ImageNet-LT.** Following (Hong et al., 2021), we apply SGD with batch size 128 and weight decay 0.0005 to train a ResNet-50 model. We perform a cosine learning rate scheme with an initial learning rate of 0.05. Unless specifically stated, we perform training for 180 epochs. We report the results of training 90/180 epochs in Table 13.

**CelebA-5.** Following (Kim et al., 2020; Wei et al., 2022), we apply SGD with batch size 128 and weight decay 0.0002 to train a ResNet-32 model for 200 epochs. We employ the linear warm-up learning rate schedule for the first five epochs and reduce it at epochs 160 and 180 by a factor of 0.01 with an initial learning rate of 0.1.

**Reproducing baseline results.** We combine our DRA with various existing methods and compare their results with other baselines methods in long-tailed recognition. For baselines combined with DRA, we implement them based on their official codes with default parameters in their modules under our training setting for fair comparison, such as model architecture and optimization hyperparameters. There are a few results disagreeing with original numbers reported in their papers, caused by their different training settings from ours, e.g. we get 44.34 top-1 accuracy of PC softmax on CIFAR100-LT while the reported number is 45.3 in (Hong et al., 2021), which is because the baseline implemented by us is not so strong as them i.e top-1 accuracy of ERM 40.28 vs 40.1 and they only perform experiments under one random seed. For other baselines, we reproduce the results using publicly available official codes or direct use numbers reported in the original papers. For all results we implement or reproduce based on official code, we run five times with different seeds and report the average.

**Study for saddle-point issue.** We use the official code[2] of (Rangwani et al., 2022a) to estimate $\lambda_{min}$ and $\lambda_{max}$, eigenvalues of Hessian by calculating the Eigen Spectral Density (Ghorbani et al., 2019). The Eigen Spectral Density is calculated on the average loss of training instances from a specific class. Specially, (Rangwani et al., 2022a) uses the PyHessian library, which uses Lanczos algorithm to compute the complete Hessian eigenvalue density efficiently. $\lambda_{min}$ and $\lambda_{max}$ are obtained from the complete Hessian eigenvalue density.

## C.2   Implementation details of DRA

The data augmentation process of DRA is summarized in Algorithm 1 while the prior work WRM (Sinha et al., 2017) is illustrated by Algorithm 2 for reference.

**Candidate value of class-wise multiplier.** Noting that the class-ware multiplier $\lambda_j$ has a negative correlation with the radius of uncertainty set so a positive correlation with the number of instances of class $j$, thus we set $\lambda_j = normalize\{N_j^{\beta}\} * S$ with $\beta, S \geq 0$ and $normalize$ meaning scale the vector to unit vector. $\beta$ determines robustness difference over classes($\beta$=0 reduces to the class-consistent $\lambda$ as in Algorithm 2) and $S$ determines overall robustness level. In this way, we could reduce the multipliers of DRA to only two hyperparameters. We set $M = 10$ and $\alpha_{inner} = 0.1$ in DRA. As for the hyperparameter $k$, the number of iterations starting to be used to augment examples, we set it as half of the whole number of iterations $M$ heuristically or $M - 1$, which is equivalent to directly utilizing WRM as the solution of our primal problem. We do not fine-tune $k$ to get an optimal number but only select from the above two choices, which, however, gets pretty good result.

**Using stage of DRA.** Inspired by (Kim et al., 2020; Samuel & Chechik, 2021), we apply DRA in the later stage of training since we need an initial model that has fit the data distribution for DRA to generate augmentation examples. Specifically, for CIFAR10-LT, CIFAR100-LT and CelebA-5 we start to use DRA

---

[2]`https://github.com/val-iisc/Saddle-LongTail`

---

**Algorithm 1** DRA: augmenting with a sequence of examples

---

1: **Input:** $\{(x_i, y_i)_{i=1}^{N_{batch}}\}$, a batch of instances
2: **Input:** $\{\lambda_j\}_{j \in [L]}$, the class-aware multipliers
3: **Output:** $Augbatch = \{(x_i^{aug}, y_i)\}_{i=1}^{N_{batch} \cdot (M-k)}$
4: $Augbatch \leftarrow \{\}$
5: **for** $i = 1, \ldots, N_{batch}$ **do**
6:    $x_i^0 \leftarrow x_i, \, l \leftarrow 0, \, g_i^* \leftarrow (2\epsilon, 0, \cdots, 0)$
7:    **while** $l < M$ and $\|g_i^*\| > \epsilon$ **do**
8:       $g_i^* = \nabla_x \lambda_{y_i} c((x_i^l, y_i), (x_i, y_i)) - l(f_\theta(x_i^l), y_i)$ {Compute gradient of inner-optimization}
9:       $x_i^{l+1} \leftarrow x_i^l - \alpha_{inner} \cdot g_i^*$ {Update augmentation examples by gradient}
10:       **if** $l \geq k$ **then**
11:          $Augbatch = concate((Augbatch; (x_i^l, y_i)))$ {Save a sequence of examples}
12:       **end if**
13:       $l = l + 1$
14:    **end while**
15: **end for**
16: **return** $Augbatch$ {Return augmented instances for outer-optimization}

---

**Algorithm 2** WRM (Sinha et al., 2017): augmenting with the last example

---

1: **Input:** $\{(x_i, y_i)_{i=1}^{N_{batch}}\}$, a batch of instances
2: **Input:** $\{\lambda\}$, the multiplier
3: **Output:** $Augbatch = \{(x_i^{aug}, y_i)\}_{i=1}^{N_{batch}}$
4: $Augbatch \leftarrow \{\}$
5: **for** $i = 1, \ldots, N_{batch}$ **do**
6:    $x_i^0 \leftarrow x_i, \, l \leftarrow 0, \, g_i^* \leftarrow (2\epsilon, 0, \cdots, 0)$
7:    **while** $l < M$ and $\|g_i^*\| > \epsilon$ **do**
8:       $g_i^* = \nabla_x \lambda c((x_i^l, y_i), (x_i, y_i)) - l(f_\theta(x_i^l), y_i)$ {Compute gradient of inner-optimization}
9:       $x_i^{l+1} \leftarrow x_i^l - \alpha_{inner} \cdot g_i^*$ {Update augmentation examples by gradient}
10:       $l = l + 1$
11:    **end while**
12:    $Augbatch = concate((Augbatch; (x_i^l, y_i)))$ {Save the last example}
13: **end for**
14: **return** $Augbatch$ {Return augmented instances for outer-optimization}

---

on 160 epochs, for Tiny-ImageNet-LT we start to use DRA on 90 epochs. For ImageNet-LT we start to use DRA on 80 epochs when training for 90 epochs while we start to use DRA on 150 epochs when training for 180 epochs. For methods that trained for 300 epochs, DRA is started to be used on 240 epochs. And for CRT, we only use DRA in the training of the second stage.

**Setting of hyperparameters.** Following Hong et al. (2021); Kim et al. (2020), we tune the hyperparameters of DRA by grid search on the validation set. WRM uses the following strategy to determine the hyperparameter multiplier: $\lambda = C \cdot E_{P_N(x)}[\|x\|_2]$ while $C = 0.04$ is a constant and $E_{P_N(x)}[\|x\|_2]$ is the average norm of all instances in training set. However, this strategy is not helpful in determining the hyperparameters $S$. For example, $E_{P_N(x)}[\|x\|_2] = 248.39$ on CIFAR10-LT while it is $222.42$ on CIFAR100-LT. The two numbers are similar but the optimal $S$ found by our grid search on the two sets disagrees a lot. Moreover, scaling the optimal $S$ on CIFAR10-LT by the proportion of $E_{P_N(x)}[\|x\|_2]$ does not work well either. How to determine the hyperparameters for DRA more elegantly could be a future research point to further improve our method.

## C.3 Full results on Tiny ImageNet-LT and ImageNet-LT

The experimental results on Tiny ImageNet-LT and ImageNet-LT are shown in Table12 and Table 13 respectively.

Table 12: Comparison of top-1 accuracy (%) on Tiny-ImageNet-LT. † means results from original papers, ‡ means reproduced results from official codes, * means requiring more training epochs. Best results are in bold and small red font denotes performance gain.

| Method | Tiny-ImageNet-LT |
|---|---|
| ERM | 34.20 |
| CE-DRS | 36.02 |
| LDAM-DRS | 37.66 |
| GIT† | 21.99 |
| LDAM-DRW | 37.43 |
| MFW*† | 35.4 |
| CDT† | 37.9 |
| IB‡ | **40.40** |
| **CE-DRA** | 35.11+0.91 |
| **CE-DRS-DRA** | 36.18+0.12 |
| **LDAM-DRS-DRA** | 39.07+1.41 |

## C.4 Full results on CIFAR10/100-LT

The experimental results and comparisons with prior methods on CIFAR10/100-LT are shown in Table 14. See Sec 5.1 for reference of methods compared.

## C.5 DRA on confidence calibration

Inspired by the discovery in our empirical study that confidence calibration is effected by both imbalance and shift of CCDs, we expect DRA would relieve over-confidence of models. From our experiments, it is kind of surprising that DRA not only improves calibration but also improves the well-calibrated models by Mixup most of the time. We use ECE (Guo et al., 2017) as the measure.

In Figure 13, it seems just with re-balancing methods model cannot be calibrated well and DRA actually gets smaller ECE on these methods, which validates again that confidence calibration is affected by shift of CCDs and DRA improves it by making the model more robust to the shift.

Table 13: The full performances on ImageNet-LT. The small red font denotes performance gain. ‡ means results from the original paper.

| Method | Many | Medium | Few | **All** |
|---|---|---|---|---|
| *90epochs* | | | | |
| ERM | **65.1** | 35.7 | 6.6 | 43.1 |
| LADE | 60.34 | 47.37 | 27.82 | 49.20 |
| Auto balance‡ | - | - | - | 49.09 |
| CRT | 61.22 | 47.52 | 26.41 | 49.52 |
| **+Ours** | 61.63 | 47.53 | **30.61** | **50.25** +0.72 |
| PC softmax | 60.4 | 46.7 | 23.8 | 48.9 |
| **+Ours** | 60.38 | 46.81 | 24.23 | 49.11+0.21 |
| Logits Adjustment | 60.62 | 47.33 | 27.53 | 49.25 |
| **+Ours** | 60.63 | **47.84** | 27.60 | 49.49+0.24 |
| *180epochs* | | | | |
| ERM | **66.84** | 40.89 | 11.54 | 46.05 |
| LADE | 62.80 | 49.76 | 33.4 | 52.14 |
| CRT | 61.59 | 47.70 | 30.32 | 50.3 |
| **+Ours** | 61.81 | 49.43 | 31.57 | 51.37+1.07 |
| PC softmax | 62.13 | 49.25 | 30.51 | 51.18 |
| **+Ours** | 62.41 | **49.72** | 31.89 | 51.64+0.46 |
| Logits Adjustment | 62.70 | 48.81 | 31.62 | 51.82 |
| **+Ours** | 63.09 | 49.58 | **32.17** | **52.42**+0.60 |

Table 14: Comparison of top-1 accuracy (%) on CIFAR10/100-LT. † means results from original papers, ‡ means reproduced results from official codes, * means requiring more training epochs. Best results are in bold and small red font denotes performance gain.

| Method | CIFAR10-LT | CIFAR100-LT | Method | CIFAR10-LT | CIFAR100-LT |
|---|---|---|---|---|---|
| **Under the setting of Hong et al. (2021)** | | | **Under the setting of Zhou et al. (2022)** | | |
| ERM | 73.24 | 40.28 | ERM | 70.45 | 38.21 |
| LDAM-DRW | 78.67 | 45.05 | M2m-LDAM‡ | 77.50 | 42.20 |
| LADE‡ | 79.87 | 45.39 | Vector loss‡ | **80.77** | 42.15 |
| **Ours** | 75.08+1.84 | 41.37+1.09 | Auto-balance† | 78.85 | 43.30 |
| | | | GIT-LDAM‡ | 78.49 | 43.49 |
| | | | Open sampling† | 79.05 | 42.86 |
| CE-DRS | 77.21 | 44.16 | CE-DRS | 75.87 | 41.21 |
| **+Ours** | 78.89+1.68 | 44.87+0.71 | **+Ours** | 77.38+1.51 | 42.61+1.40 |
| LDAM-DRS | 78.71 | 45.12 | LDAM-DRS | 77.47 | 42.78 |
| **+Ours** | 79.75+1.04 | 45.54+0.42 | **+Ours** | 78.76+1.29 | **43.53**+0.77 |
| PC softmax | 79.58 | 44.34 | PC softmax | 77.92 | 41.86 |
| **+Ours** | **80.39**+0.81 | 44.84+0.50 | **+Ours** | 79.28+1.36 | 43.44+1.59 |
| Logit Adjustment | 79.65 | 45.17 | Logit Adjustment | 78.07 | 42.08 |
| **+Ours** | 80.33+0.68 | **45.71**+0.54 | **+Ours** | 79.13+1.06 | 42.96+0.88 |

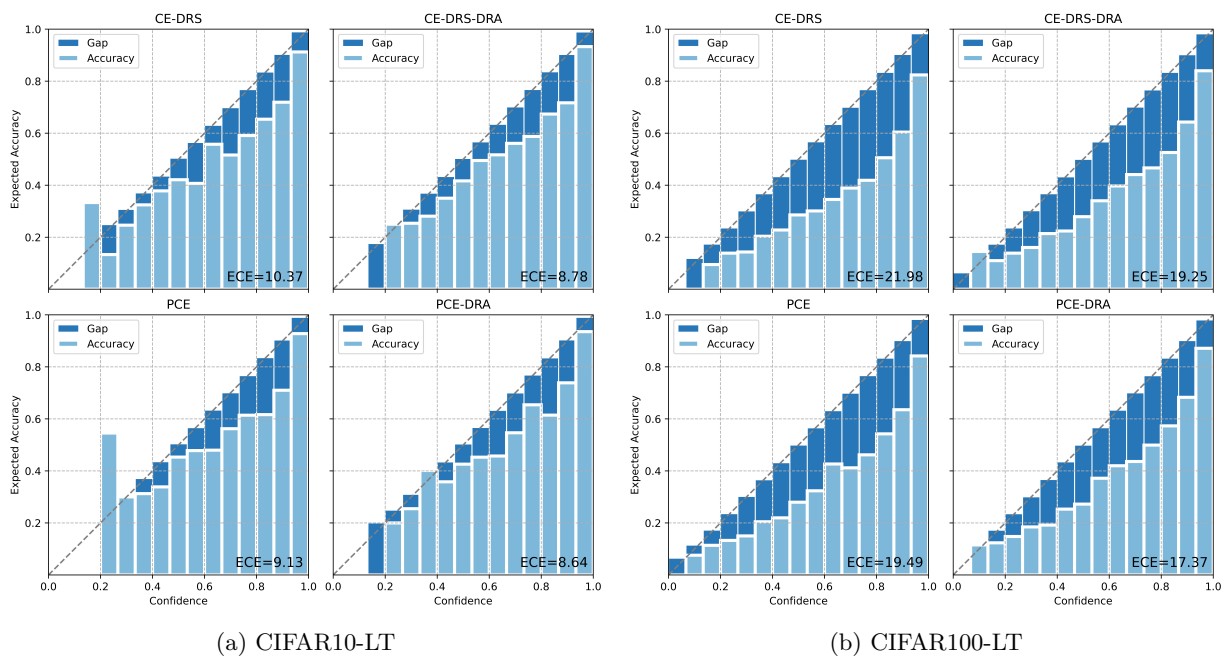

(a) CIFAR10-LT                      (b) CIFAR100-LT

Figure 13: ECE (%) and reliability diagram on CIFAR10-LT/CIFAR100-LT. DRA gets better calibration performance marginally. PCE means PC softmax.

Besides, considering the discovery in recent works (Zhong et al., 2021; Xu et al., 2021) that Mixup has a significant positive effect on calibration, we conduct experiments to measure ECE with both Mixup and DRA. Figure 14 shows DRA further improves calibration on the basis of Mixup most of time. The only exception is CE-DRS on CIFAR10-LT, in which DRA weakens calibration but boosts its accuracy as shown in Table 2. It seems with both Mixup and DRA the regularization on logits is too strong to obtain good calibration as the model gives confidence even lower than the accuracy instead of usual over-confidence on neural network (Guo et al., 2017). These results also raise an interesting question: Do calibration and accuracy agree in long-tailed recognition? Or more specifically, does good calibration mean a good model in long-tailed recognition (Xu et al., 2021)? Actually, our experiment shows that it is possible to boost performance while weakening calibration.

### C.6 Visualization of and discussion on examples generated by DRA

To get more precise understanding, we visualize the examples generated by DRA, as in Figure 15. It exhibits that DRA adds pixel-wise transformation to instances in order to generate harder examples as data augmentation. Some of these transformations tend to erase conspicuous features in an instance by adding "noise". In this way, model is encouraged to extract broader features instead of overfitting features it has learned. For example, DRA seems to add "noise" to erase the ladder from "fire engine" in the first row. As ladder is a salient feature for fire engine, and this augmentation makes model attend to more recognizable features e.g. water tank and red color. Similarly, in the second row, DRA adds "noise" on the beaks and claws of the bird and encourages the model to take in more features to help recognition instead of over-depending on the two features.

In addition, more surprisingly, DRA can catch semantic information while discard nuisance information by itself. As in the fourth row, the yellow sweet peppers happen to be transformed to green while pixels out of the peppers almost kept unchanged. In other words, DRA makes a transformation of color, and the transformed instance still belongs to "sweet pepper" class. In this way, DRA performs transformations that keep the semantics unchanged. As shown in the bottom two rows, DRA could give transformations on the background (nuisance information) and keep semantic information: almost all of the pixel-wise transformations in the

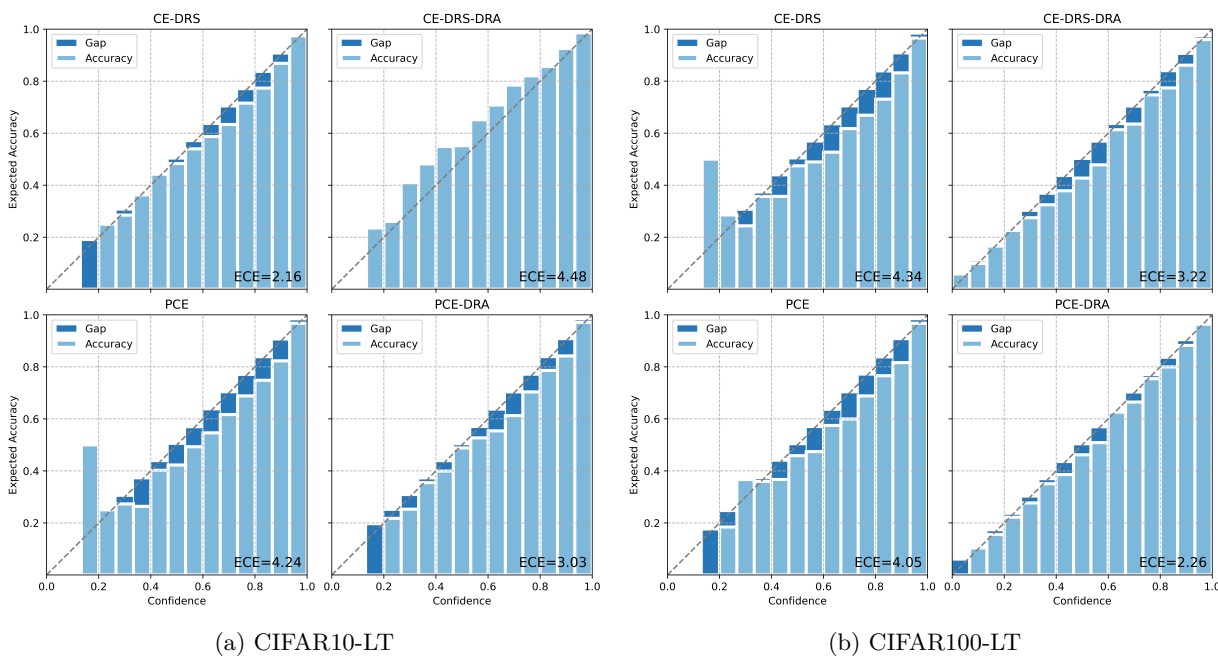

(a) CIFAR10-LT        (b) CIFAR100-LT

Figure 14: ECE (%) and reliability diagram on CIFAR10-LT/CIFAR100-LT with Mixup. DRA even improves the calibration of models with Mixup, which have obtained small ECE and been calibrated well. PCE means PC softmax.

last but one row lie in the sky, and the semantic object "road" isn't changed. In the last row, the pixels of sofa are not transformed while the background is changed.

From these results, we infer that seeking harder examples for a robust model sometimes agrees with transformations that keep the semantics of instances. For example, background intensity could serve as examples to obtain distributional robustness under a few situations, and this can be an explanation of why GIT improves performance by adding transformations keeping semantics of instances. However, when transformations that keep semantics are not that effective to gain distributionally robustness, DRA tends to seek other examples with obvious features are erased.

From the observation that examples from DRA do not have to keep semantics and seem like instances with "noise", we hope DRA could suggest a new motivation for data augmentation: making a more robust model for unknown distribution shift instead of utilizing information to reduce the shift, e.g. GIT, ISDA (Zhou et al., 2022; Wang et al., 2019; Li et al., 2021). This motivation, making a model robust to shift, could give hints to explain the rationality of existing data augmentations that don't have to keep semantics, e.g. adding pure noise to instances as data augmentation (Zada et al., 2022) and those based on Mixup (Zhang et al., 2017; Chou et al., 2020; Zhong et al., 2021; Xu et al., 2021).

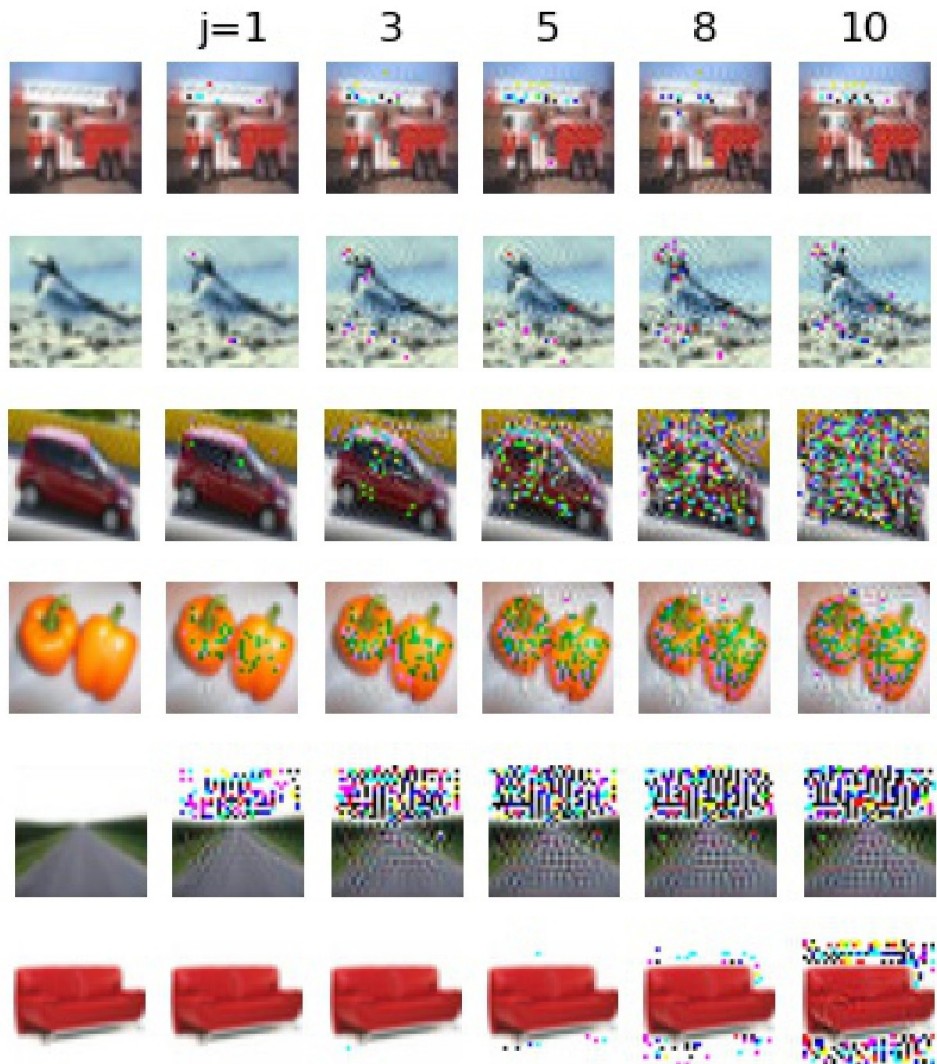

Figure 15: Examples generated by DRA on CIFAR-LT. Left most column are original examples and $j$ is the number of iterations. DRA tends to add pixel-level transformations to obtain harder examples while some of these transformations agree with semantic information.

