# OpenReview forum: "Enhancing Robustness to Class-Conditional Distribution Shift in Long-Tailed Recognition"
_TMLR — Accepted by TMLR_

### Review · Reviewer_gVMi · 2023-11-18

**Summary Of Contributions:**

This paper proposes a data augmentation method termed Distributionally Robust Augmentation (DRA) to enhance the robustness of models to class-conditional distribution (CCD) shift in long-tailed recognition. The paper conducts a thorough empirical study on the impact of CCD shift on existing re-balancing and data augmentation methods, and shows that CCD shift is a critical factor that limits the performance of long-tailed recognition, especially for tail classes. The paper then formulates a min-max optimization problem to train models that are robust to a class-aware uncertainty set of potential distributions. The paper also presents an efficient algorithm to implement DRA by generating a sequence of augmented examples in the inner optimization loop.

**Audience:**

Yes

**Broader Impact Concerns:**

What are the possible ethical and social implications of the proposed method for long-tailed recognition, especially for the underrepresented or marginalized groups that correspond to the tail classes? How can the method ensure fairness, accountability, and transparency in its decision making?

**Claims And Evidence:**

Yes

**Requested Changes:**

The primary questions for the rebuttal primarily arise from the "weaknesses" section.

**Strengths And Weaknesses:**

**Paper Strength**

- The paper addresses an important and challenging problem of long-tailed recognition, and provides a comprehensive analysis of the CCD shift issue from both empirical and theoretical perspectives.
- The paper conducts extensive experiments on five long-tailed classification benchmarks, and shows that DRA can significantly improve the performance of existing re-balancing and data augmentation methods, as well as alleviate some issues such as confidence calibration and saddle-point.

**Paper Weakness**

- How does the proposed method handle the uncertainty and noise in the data, especially for the tail classes that have scarce and unreliable samples?
- **Lack of ablation study on the choice of cost function.** The paper uses a specific cost function $c$ for the DRO problem, but does not provide any justification or analysis for these choices. It is unclear how sensitive the DRA method is to different choices of $c$. A more thorough ablation study on the DRA method would be helpful.
- **Lack of clarity and details in the theoretical analysis.** The paper provides a generalization bound for the DRA method, but does not explain the meaning and derivation of some terms and conditions in the bound, such as $C(N_j)$.

---

> ### Author Response · Authors · 2023-12-29
> **Response to Reviewer gVMi(1/2)**
>
> Thanks for your the careful and constructive comments and we hope our clarification will address your concerns.
>
>  > How does the proposed method handle the uncertainty and noise in the data
>
> Actually, the focus of this paper i.e. CCD shift from scarcity of instances, to some extent, comes from the uncertainty of data. As the number of instances from tail-classes is limited, they are more likely to be affected by the randomness of data collection, causing probably biased estimation of the ideal test distribution and bad performance, as stated in Sec 3.
>
> The proposed DRA aims at training models robust to the CCD shift, by considering the worst risk on a set of all potential distributions within a certain discrepancy from empirical distribution of collected samples from a class. By adjusting the pre-class discrepancy, DRA covers the risk on ideal test distribution, regardless of the scarcity and uncertainty of data, as the bound (7) indicates.
>
> In this work, we assume that there is no noise on the label or the instance i.e. for each instance $x_i$ labeled $y_i$, $x_i$ is sampled from the real(ideal) class-conditional distribution $P(x|y_i)$, as the setting of most prior works[1,2,3]. We highlight the potential of improving data scarcity issue under noise data in the long-tailed recognition as they may co-occur in the real-world long-tailed distribution.
>
>  > Selection of cost function
> 1. __Why choose square of L2 distance?__ We typically select $c_{x}(x_1,x_2)=\|x_1-x_2\|_2^2$ ( $\|x_1-x_2\|_2$ is a typo), because it naturally satisfies all the assumptions for our theoretical analysis (nonnegative, lower-semi-continuous, etc, for a valid cost function, the transformation to min-max optimization(Theorem 7 in the new version) and the generalization (Theorem 14 in the new version), and 1-strongly convex for the convergence(Theorem 9 in the new version)). For better clarify, in the new version, we specify the assumption on the cost function in each theorem and $c(\cdot, \cdot)=\|x_1-x_2\|_2^2+\infty\cdot\mathbb{1}\{y_1\neq y_2\}$ is a typical example to meet all the assumptions of them.
> 2. __Abalation study of other choices on cost function.__
> Other than our typical choice of $c(\cdot, \cdot)=c_{x}(x_1,x_2)+\infty\cdot\mathbb{1}\{y_1\neq y_2\}$ with $c_{x}(x_1,x_2)=\|x_1-x_2\|_2^2$, following your constructive comments, we add ablation study on the cost function, with other choices of $c_x$ in $\{\|x_1-x_2\|_p^r|p=1,2,4,r=1,2\}$ as in the prior work WRM[4].
>
> |(Baseline:CE-DRS):77.21|CIFAR10-LT|p=1|2|4|
> |--|--|--|--|--|
> |r=2|k=M-1|nan|78.89|77.78|
> |-|k=M/2|nan|77.83|78.56|
> |r=1|k=M-1|56.78|73.24|77.34|
> |-|k=M/2|79.12|78.07|77.41|
> |(Baseline:CE-DRS):44.16|CIFAR100-LT|P=1|2|4|$\infty$|
> |r=2|k=M-1|nan|44.57|44.45|
> |-|k=M/2|nan|44.87|44.31|
> |r=1|k=M-1|40.57|44.64|44.72|
> |-|k=M/2|44.81|45.12|44.75|
>
> However, these functions are not 1-strongly convex so fail to apply our convergence guarantee like the case of the typical cost with small $\lambda_j$, which causes unstable inner optimization(Sec 4.2). For these cost functions, we indeed observe that the inner optimization becomes unstable and may cause lower performance than baseline or even diverge. In these cases, the sequence-augmented data(k=M/2) effectively alleviates this issue and gains consistent improvement over baseline, serving as appropriate augments for only the last example(k=M-1).
>
> Following your and Reviewer CjJm's advice, we also conduct more ablation study on our proposed DRA. Please
> refer to our response to Reviewer CjJm or Sec 5.3 in our updated version.

---

> > ### Author Response · Authors · 2023-12-29
> > **Response to Reviewer gVMi(2/2)**
> >
> > > Clarity and details of the generalization bound
> >
> > For fluency and conciseness, we put the formal version and detailed derivation of the bound in the Appendix A.4. And we have added more clarification of the bound in the body of the new version.
> >
> >  > Potential ethical concerns
> >
> > In this work, the setting of long-tailed recognition follows a series of prior works [1,2,3], which assume that there exists a training distribution with imbalanced label distribution and models are evaluated on a balanced test distribution, so there are no extra ethical concerns brought by this work. Generally, there can be potential negative impacts associated with long-tailed recognition e.g. privacy threats in face recognition since face data often exhibits long-tailed distribution over entities or objects [4].
> >
> > For the tail classes that correspond to the underrepresented or marginalized groups, this work shows that unreliable distribution estimation from scarce data limits models to make accurate predictions for these classes. The proposed DRA trains model robust to unreliable distribution estimation, thus avoids potential bias or prejudice to some extent.
> >
> > Within the scope of long-tailed visual recognition, commonly used neural network models exhibit a black-box style, making it difficult to provide other explanations for their predictions besides the corresponding output logits as confidence. And the confidence calibration, i.e. how much the reference logits estimate the correctness of model's prediction, is usually bad for neural networks [6], which is even worse under the long-tailed distribution [7]. Our proposed DRA not only improves prediction accuracy but also improves confidence calibration, enhancing transparency and reliability over previous methods.
> >
> > [1] Long-tail learning via logit adjustment. ICLR 2021.
> > [2] Learning Imbalanced Datasets with Label-Distribution-Aware Margin Loss. NeurIPS 2019.
> > [3] Disentangling Label Distribution for Long-tailed Visual Recognition. CVPR 2021.
> > [4] Range loss for deep face recognition with long-tailed training data. CVPR 2017.
> > [5] Certifying Some Distributional Robustness with Principled Adversarial Training. ICLR 2018.
> > [6] On Calibration of Modern Neural Networks. ICML 2017.
> > [7] Improving Calibration for Long-Tailed Recognition. CVPR 2021.

---

### Review · Reviewer_CjJm · 2023-11-20

**Summary Of Contributions:**

This paper first conducts a series of experiments to study the adverse impact of the Class-Conditional Distribution (CCD) shift on the long-tailed recognition problem. Then, the authors propose Distributionally Robust Augmentation (DRA) to make models robust to CCD shift. Meanwhile, some theoretical analyses are provided to prove its rationality. Finally, empirical results demonstrate the effectiveness of DRA on long-tailed classification tasks.

**Audience:**

Yes

**Broader Impact Concerns:**

There are no concerns about the ethical implications.

**Claims And Evidence:**

Yes

**Requested Changes:**

- Additional experiments to study the effectiveness of class-aware radius and sequence augmented data might strengthen the work.

-  Compare and review a broader range of SOTA methods.
For example, they might consider methods like

- "Self-Supervised Aggregation of Diverse Experts for Test-Agnostic Long-Tailed Recognition" ,
- "A Unified Generalization Analysis of Re-Weighting and Logit-Adjustment for Imbalanced Learning",
- "Class-Conditional Sharpness-Aware Minimization for Deep Long-Tailed Recognition",
- "Learning with Multiclass AUC: Theory and Algorithms"

**Strengths And Weaknesses:**

Strengths:

- The experiments and analyses to investigate the adverse impact of CCD shift are comprehensive. Specifically,  the authors employ various methods under settings with or without removing shift sampling to show that CCD shift is a key factor that limits the performance. Then, the authors take a step further and reveal that CCD shift hinders re-balancing methods in feature learning and causes sub-optimality of Fisher-consistent post-hoc adjustment. Additionally,  the authors also demonstrate that CCD shift affects confidence calibration and feature deviation. Through these thorough experiments,  they effectively illustrate the adverse effects of CCD shift.
- This paper offers numerous theoretical analyses to demonstrate DRA's rationality and solidity. Firstly, the authors convert Distributional Robust Optimization (DRO) into a min-max optimization problem and present the objective function.  The authors introduce a theorem to prove that balanced risk can be bounded by the objective. Furthermore, they propose DRA, using a sequence of examples to enhance DRO and make the optimization more stable. They further provide a theorem to explain the rationality. These theoretical analyses makes the proposition of DRA reliable.
- Numerical results indicate that DRA is effective. Specifically, empirical results show that DRA can enhance different re-balancing methods and can cooperate with other data augmentation to further improve accuracy. What's more, the experiments also demonstrate that DRA can improve confidence calibration and alleviate the saddle-point issue.



Weakness:
- The authors solely conduct experiments comparing "CE-DRS-SRA ($\beta=0.30, k=M-1$)" with "WRM ($\beta=0,k=M-1$)" to investigate the effectiveness of class-aware radius, which might not be sufficient. Similarly, the investigation of the utilization of sequence-augmented data is also limited.
- There are some typos. For example,
  - In section 4.3, "which is a derived issue of CCD shift, as found in our study (Sec 3.3)" should be "which is a derived issue of CCD shift, as found in our study (Sec 3.3)"

---

> ### Author Response · Authors · 2023-12-29
> **Response to Reviewer CjJm**
>
> Thanks for your the careful and constructive comments and we hope our clarification will address your concerns.
>  > More ablation study on class-aware radius and sequence-augmented data
>
> Following your and Reviewer gVMi's advice, we conduct more ablation study on our proposed DRA
>
> 1. __On other cost functions.__
> Other than our typical cost function $ c(\cdot, \cdot)=c_{x}(x_1,x_2)+\infty\cdot\mathbb{1}\{y_1\neq y_2\} $ with $c_{x}(x_1,x_2)=\|x_1-x_2\|_2^2$, we conduct experiments on other $c_x\in \{ \|x_1-x_2\|_p^r|p=1,2,4,r=1,2 \} $.
>
> |(Baseline:CE-DRS):77.21|CIFAR10-LT|p=1|2|4|
> |--|--|--|--|--|
> |r=2|k=M-1|nan|78.89|77.78|
> |-|k=M/2|nan|77.83|78.56|
> |r=1|k=M-1|56.78|73.24|77.34|
> |-|k=M/2|79.12|78.07|77.41|
> |(Baseline:CE-DRS):44.16|CIFAR100-LT|P=1|2|4|$\infty$|
> |r=2|k=M-1|nan|44.57|44.45|
> |-|k=M/2|nan|44.87|44.31|
> |r=1|k=M-1|40.57|44.64|44.72|
> |-|k=M/2|44.81|45.12|44.75|
>
> However, these functions are not 1-strongly convex so fail to apply our convergence guarantee like the case of the typical cost with small $\lambda_j$. For these cost functions, we indeed observe that the inner optimization becomes unstable and may cause lower performance than baseline or even diverge. In these cases, the squence-augmented data(k=M/2) effectively alleviates this issue and gain consistent improvement over baseline, serving as appropriate augments for only the last example(k=M-1).
>
> 2. A sequence or a single example besides the last one.
>
> The proposed sequence-augmented data strategy is mainly to alleviate the instability of only using last example in some cases.
> Moreover, we probe if the effectiveness of our strategy is from using a sequence or simply from examples not the last one on CIFAR100-LT. Specifically, we compare a variant of inner optimization of randomly returning an example within the last M/2 iteration, noted as 'random example'.
>
> |S($\beta$=0.30)|60|70|80|90|100|
> |-|-|-|-|-|-|
> k=M/2|44.79|44.87|44.69|44.53|44.48
> k=M-1|44.37|44.45|44.50|44.57|44.48
> random single example|44.28|44.58|44.74|44.47|44.37
>
> The result of this variant under the same setting of our ablation study is shown above, which is worse than a sequence of examples most of the time but overperforms the last example under smaller S, supporting our insight reflected by (12) that only using the last example is not stable for optimization.
>
> 3. More ablation study on class-aware radius
>
> CE-DRS(CIFAR10-LT)|$\beta$=0.15|0.20|0.25|0.30|0.35|0.4|0(WRM)
> |-|-|-|-|-|-|-|-|
> S=100|77.75|78.35|78.31|78.46|78.89|78.74|77.31|
> S=70|77.41|77.46|77.74|78.35|78.24|78.43|77.39
> CE-DRS(CIFAR100-LT)|$\beta$=0.15|0.20|0.25|0.30|0.35|0.4|0(WRM)
> S=100|44.29|44.57|44.59|44.48|44.65|44.46|44.16
> S=70|44.40|44.34|44.74|44.87|44.81|44.85|44.23|
>
> We also add abaltion study on class-aware radius by adjusting $\beta$ to control the radius of the uncertainty set for every class. Our class-aware radius shows improvement over WRM($\beta$=0). Moreover, the $\beta$ needs to be searched widely to get the best performance, indicating that robustness for CCD shift is essential for long-tailed recognition and needs to be assigned carefully over classes.
>
>   > Missed baselines and references
>
> We appreciate the reminder of missed baselines and references. We have added reviews and discussion for more prior works in the related work, including [1,2,3,5], and added more comparisons on performance and training cost with [4,5]. Specifically, for [5], after consulting the authors, we use the results with mixup augmentation reported in its appendix for fair comparison, instead of results with strong augmentation(auto augmentation and cutmix) in the body of the paper.
>
> [1]  Self-Supervised Aggregation of Diverse Experts for Test-Agnostic Long-Tailed Recognition. NeurIPS 2022.
> [2] A Unified Generalization Analysis of Re-Weighting and Logit-Adjustment for Imbalanced Learning.  NeurIPS 2023.
> [3] Long-tailed recognition by routing diverse distribution-aware experts. ICLR 2021.
> [4] Escaping Saddle Points for Effective Generalization on Class-Imbalanced Data. NeurIPS 2022.
> [5] Class-Conditional Sharpness-Aware Minimization for Deep Long-Tailed Recognition. CVPR 2023

---

> > ### Comment · Reviewer_CjJm · 2024-01-01
> >
> > I do appreciate the authors' feedback. I think they have addressed most of my concerns.
> >
> > I have two more suggestions before publification:
> >
> > 1)  The theoretical results should be summarized in the main paper to be self-contained
> >
> > 2) More details should be given for the proof of thm.11. I don't understand how comes "with probability 1- \eta,  P(X|y) \in P_j' be setting the specific form of radius.

---

> > > ### Author Response · Authors · 2024-01-06
> > > **Response to Reviewer CjJm**
> > >
> > > Thank you for your positive feedback and acknowledgment of our efforts to address your concerns. We are pleased to hear that most of your points have been satisfactorily resolved.
> > >
> > > We also appreciate your constructive suggestions. We have transformed the main formulas into theorem form in the body of the new submitted version accordingly, to make it self-contained. For the clarification of Theorem 11(Theorem 14 in the new version), based on the estimation of Wasserstein distance between distributions(with the probability of $1-\eta$) in Lemma 3, by specifying the radius into the form Theorem 11, the radius of P_j is greater than the distance W_{c}(P(x|y_j),P_{N,LT}(x|y_j)) and thus $P(x|y_j) \in {P}_{j}$. We also added above clarification into the proof and hope that it would clarify any ambiguities and make the proof more comprehensible.
> > >
> > > If there are any further suggestions or clarifications needed, please feel free to let us know.

---

### Review · Reviewer_v3rC · 2023-12-15

**Summary Of Contributions:**

The paper proposes a distributionally robust augmentation (DRA) to alleviate the so-called class-conditional distribution (CCD) shift in long-tail image recognition. The effect of removing CCD shift is first unveiled by an empirical study and then DRA is proposed to alleviate this CCD shift problem in long-tail image recognition. The proposed DRA generates adversarial examples by maximizing the loss function while keeping the sample close to the original one measured by $l_2$ distance. One difference between adversarial training and DRA is that DRA uses samples from the optimization trajectory while adversarial training only uses the sample of final step. They show that the proposed DRA has some benefits over baselines on some datasets.

**Audience:**

Yes

**Broader Impact Concerns:**

Not applicable.

**Claims And Evidence:**

No

**Requested Changes:**

The paper needs to go through a major revision to address the concerns and answer the questions in the **weaknesses** section.

**Strengths And Weaknesses:**

**Strengths**

$\cdot$ The long-tail image recognition problem is an important one in practice.

$\cdot$ The empirical comparison between DRA and existing baselines is solid and extensive.

**Weanesses**

1) The concept of CCD shift is not clearly explained in the paper. I understand the mathematical meaning of CCD, but the removing CCD shift experiment is quite confusing to me (see Q1 below). Since the CCD is a core concept in this paper, it should be better explained via some visualization. The paper only shows accuracy of removing CCD, but the exact concept of CCD is not properly illustrated. Specifically, it is not clearly stated that removing CCD shift in Section 3.3 brings more training data and the description in the paragraph **Implementation of removing shift sampling** is not clear.

2) The effect of DRA is claimed to mitigate CCD shift. However, this effect is not supported by experimental results. The experiment shows that some performances are improved by DRA, but the reason behind this improvement is not necessarily connected to removing CCD shift. In other words, is there any quantitative measure for the reduced CCD shift in the experiment? It would be more convincing to add such a measure.

3) The performance of DRA as shown in Table 2 cannot support its effectiveness in some cases. For instance, on ImageNet-LT, the performance of DRA is not as good as CMO in Few data setting, and not as good as MISLAS in both Few and Med setting. Could the authors add the discussion on the lower performance of DRA in such cases? In addition, in the previous version of the submission, the performance of PC Softmax and PC Softmax + Ours in ImageNet-LT Few with Mixup is 32.42 and 32.0. Why does the performance of PC Softmax and PC Softmax + Ours become 30.57 and 32.20 in this version?

**Additional comments**

Note that the three points are all discussed by the authors in the response of a previous conference, where the authors state that they **would add a discussion of above issue (the three concerns) in the next version of our manuscript**. However, those updates are not shown in this updated version.

---

> ### Author Response · Authors · 2023-12-29
> **Response to Reviewer v3rC(1/2)**
>
> Thanks for your careful and constructive comments and we hope our clarification will address your concerns.
>
>  > Clarification of removing shift sampling and CCD shift.
>
> The key of the proposed removing shift sampling is changing the CCD of original long-tailed dataset into the CCD from the balanced dataset, as formulated in (2). By sampling from the modified distribution, potential candidates of each class are from the whole balanced dataset instead of the limited number of instances in the long-tailed dataset. In this process, the training relies on the balanced dataset and more novel instances are brought and seen by the model compared with vanilla training on long-tailed datasets, while the label distribution is kept still for valid ablation.
>
> Following your constructive advice, we add above note as further clarification on removing shift sampling in Sec 3.2.  To better clarify the concept of CCD shift, we further visualize a toy example and numerical results on it in Appendix B.1, which directly validate Proposition 1 and show that scarce instance is hard to reliably estimate the risk on the distribution of the class.
>
>  > The effectiveness of DRA on CCD shift.
>
> Our primary objective with DRA is not to directly mitigate CCD shift.  As CCD shift comes from scarce instances in tail classes under long-tailed recognition setting, it is inherently challenging to be directly removed or mitigated without adding extra i.i.d. sampled data. Instead, our DRA approach aims at achieving distributional robustness to cope with CCD shift. Note that though the inner optimization of DRA can be realized as data augmentation, examples from it act as the potential worst perturbation, instead of restoring the real distribution of tail classes and removing CCD shift.
>
> Actually, we have provided multi-view evidence supporting that the distributional robustness introduced by DRA is effective in improving models trained on long-tailed distribution. (1) The empirical results of DRA show it leads to significant improvement over baselines of classification and confidence calibration, which suffer from CCD shift. (2) The ablation study (Figure 5, 6) exhibits that the setting of radius of the uncertainty set (multiplier $\lambda_j$) is essential for the performance gain from DRA, and the best accuracy is often obtained with relatively large $\beta$($>0.25$), i.e. seeking much more robustness for tail classes than head classes, which aligns with the finding in Sec. 3 that CCD shift is more notable on the classes with fewer instances. (3) An analysis of the loss landscape of solutions from DRA (Sec. 5.3) shows that DRA obtains more flat solutions in the parameter space and thus has better generalization and stability to potential distribution shift [1], further validating that DRA would obtain more robust solutions.
>
> Moreover, quantifying shift between distributions or the robustness of a specific model towards a set of distributions is difficult in practice and out of the scope of this work, since (1) metrics like the Wasserstein distance, are inherently complex and computationally expensive for high-dimensional random variables [2]. (2) quantifying relevance between the introduced distributional robustness and the specific DRO procedure or the size of uncertainty set, remains open and lacks systematic conclusions [3,4].

---

> > ### Author Response · Authors · 2023-12-29
> > **Response to Reviewer v3rC(2/2)**
> >
> > > The empirical effectiveness of DRA.
> >
> > In Table 2, DRA brings consistent improvement on overall performance over various baselines and data augmentation methods, and gains improvement on all three splits of ImageNet-LT, e.g. DRA improves CRT-Mixup on all three splits by a large margin (over 1.2%), and obtains performance gain of 1.15, 1.82 and 0.72 on three datasets when cooperating with the recent CUDA [5], which validates that DRA is effective across multiple scenarios.
> >
> > For some prior works, as you point out, DRA does not overperform them on some splits (CMO, Mislas). Though these methods obtain higher accuracy on tail classes (few split), their overall performance is worse than DRA, and the performance on head classes (many, mid splits) may be even worse than the baseline (BS-CMO). We suppose that it is the accuracy metric that causes unaligned comparative results on different splits. Note that the model selection is based on overall performance, while the accuracy is reported with respect to specific split.  In fact, we find that there exist hyperparameters of DRA that can obtain much higher few split accuracy, i.e. larger $\beta$ to favor tail classes, but get low overall accuracy, while other methods e.g. CMO may sacrifice the head classes for high overall performance. Therefore, considering various inductive bias of methods, focusing on the comparison of a single split is not comprehensive.
> >
> > A more extreme example is the backward-LT expert in SADE [6] trained by backward Balanced Softmax, which gets better performance on few split than head split regardless of overall performance. This issue is further explored by recent work [7] which shows the model obtaining the best worst-class performance usually fails to obtain the best overall performance. We add above discussion as further analysis to Sec. 4.
> >
> >  > The updated results in Table 2.
> >
> > As stated above, after last submission, we try to search for hyperparameters more finely to obtain good results in both overall performance and split performance. In this process, we find that the original few split result of PC softmax is a typo. Then we confirm it by re-running the PC softmax experiments and update Table 2 with the new results.
> >
> >  > Add a discussion of above issues.
> >
> > In the response to a previous venue, we discuss on above questions and a few further ones but do not get any acknowledgement on whether our response addresses your concerns. Therefore, we do not add the full discussion into the submission.  We add related discussions to the manuscript(clarification of removing shift in Sec. 3.2, visualization of CCD shift by toy example in Appendix B.1) and hope our response answers your questions well this time.
> >
> > [1] Escaping Saddle Points for Effective Generalization on Class-Imbalanced Data. NeurIPS 2022.
> > [2] A Wasserstein-type distance in the space of Gaussian Mixture Models. SIAM Journal on Image Science.
> > [3] Incorporating Unlabeled Data into Distributionally Robust Learning. JMLR 2021.
> > [4] Does Distributionally Robust Supervised Learning Give Robust Classifiers? ICML 2018.
> > [5] CUDA: Curriculum of Data Augmentation for Long-tailed Recognition. ICLR 2023.
> > [6] Self-Supervised Aggregation of Diverse Experts for Test-Agnostic Long-Tailed Recognition. NeurIPS 2022.
> > [7] No One Left Behind: Improving the Worst Categories in Long-Tailed Learning. CVPR 2023.

---

> > > ### Comment · Reviewer_v3rC · 2024-01-16
> > > **Thanks for the response**
> > >
> > > Thanks for the response. My major concerns are addressed and I don't object to accepting the paper.

---

### Decision · Action_Editor_xpqw · 2024-02-03

**Recommendation:** Accept as is

**Comment:**

All reviewers appreciated the soundness of the proposed DRA, as well as the extensive experiments.
The revision has addressed all the concerns of the reviewers.

**Audience:**

Yes.

**Claims And Evidence:**

Claims are properly supported.

---

> ### Author Response · Authors · 2024-02-22
>
> The camera-ready version of this paper has now been submitted, and we sincerely thank the action editors and the reviewers for their insightful comments and guidance throughout the revision process. Your expertise has been invaluable in bringing our manuscript to its final form.